EMBO
Molecular Medicine

# B cell lineage reconstitution underlies CAR-T cell therapeutic efficacy in patients with refractory myasthenia gravis

Dai-Shi Tian[1,2,6], Chuan Qin[1,2,6], Ming-Hao Dong[1,2,6], Michael Heming [ID][3], Luo-Qi Zhou[1,2], Wen Wang [ID][4], Song-Bai Cai [ID][4], Yun-Fan You[1,2], Ke Shang[1,2], Jun Xiao[1,2], Di Wang[5], Chun-Rui Li[5], Min Zhang[1,2], Bi-Tao Bu[1,2], Gerd Meyer zu Hörste [ID][3✉] & Wei Wang [ID][1,2✉]

## Abstract

B-cell maturation antigen (BCMA), expressed in plasmablasts and plasma cells, could serve as a promising therapeutic target for autoimmune diseases. We reported here chimeric antigen receptor (CAR) T cells targeting BCMA in two patients with highly relapsed and refractory myasthenia gravis (one with AChR-IgG, and one with MuSk-IgG). Both patients exhibited favorable safety profiles and persistent clinical improvements over 18 months. Reconstitution of B-cell lineages with sustained reduced pathogenic autoantibodies might underlie the therapeutic efficacy. To identify the possible mechanisms underlying the therapeutic efficacy of CAR-T cells in these patients, longitudinal single-cell RNA and TCR sequencing was conducted on serial blood samples post infusion as well as their matching infusion products. By tracking the temporal evolution of CAR-T phenotypes, we demonstrated that proliferating cytotoxic-like CD8 clones were the main effectors in autoimmunity, whereas compromised cytotoxic and proliferation signature and profound mitochondrial dysfunction in CD8[+] Te cells before infusion and subsequently defect CAR-T cells after manufacture might explain their characteristics in these patients. Our findings may guide future studies to improve CAR T-cell immunotherapy in autoimmune diseases.

**Keywords** Chimeric Antigen Receptor (CAR) T-cell Immunotherapy; Refractory Myasthenia Gravis; B Cell Maturation Antigen; Single-Cell RNA Sequencing
**Subject Categories** Immunology; Musculoskeletal System

## Introduction

Myasthenia gravis (MG) is a heterogenous autoimmune disease characterized by muscle weakness and fatigue. Autoreactive B cells with autoantibody formation play a key role in the pathogenesis of MG, and the autoantibodies in the majority of MG patients are directed against the muscle acetylcholine receptor (AChR) of the neuromuscular junction, while in 6% antibodies against the muscle-specific kinase (MuSK) are detected (Gilhus, 2016; Gilhus and Verschuuren, 2015). B-cell maturation antigen (BCMA), primarily expressed in plasmablasts and plasma cells (PB/PCs) at high levels (Mikkilineni and Kochenderfer, 2021), represents a promising target antigen for novel therapies in autoantibody seropositive MG. Considering the pathophysiology of MG, treatment with chimeric antigen receptor (CAR) T cells that recognize BCMA[+] B cells might be useful in refractory forms of the disease (Verschuuren et al, 2022). Here we present the cases of two refractory MG patients, one with AChR-IgG and the other with MuSK-IgG, who received anti-BCMA CAR T cells (ClinicalTrials.gov, number NCT04561557). In order to investigate the evolutionary paths of T cells derived from patients and CAR-T cells in infusion products (IPs), as well as their behavior after transfer in vivo, we conducted single-cell transcriptomic and T-cell receptor (TCR) sequencing analyses on samples obtained from the two patients pre- and post treatment. By clone tracking, we observed a shift in T-cell lineages from autologous T effector (Te) cells towards proliferating CAR-T cells in IPs, and cytotoxic and natural killer (NK)-like Te cells post treatment in vivo. Through integrating our sequencing data with published datasets, we demonstrated that the compromised cytotoxic properties and enhanced mitochondrial dysfunction of the autologous Te cells in MG patients, might underly the characteristics of CAR-T cells in autoimmunity.

[1]Department of Neurology, Tongji Hospital, Tongji Medical College, Huazhong University of Science and Technology, 430030 Wuhan, China. [2]Hubei Key Laboratory of Neural Injury and Functional Reconstruction, Huazhong University of Science and Technology, 430030 Wuhan, China. [3]Department of Neurology with Institute of Translational Neurology, University Hospital Münster, Münster, Germany. [4]Nanjing IASO Biotechnology Co., Ltd, 210018 Nanjing, China. [5]Department of Hematology, Tongji Hospital of Tongji Medical College, Huazhong University of Science and Technology, 430030 Wuhan, China. [6]These authors contributed equally: Dai-Shi Tian, Chuan Qin, Ming-Hao Dong.
✉E-mail: gerd.meyerzuhoerste@ukmuenster.de; wwang@tjh.tjmu.edu.cn

# Results

## Patients' characteristics and clinical response

**MG-1** was a 33-year-old woman with typical AChR-IgG and Titin-IgG seropositive generalized early-onset MG. Thymectomy had been performed at disease onset (21 months before study enrollment), with a pathological diagnosis of type AB thymoma. Clinical remission was not achieved under regular administration of acetylcholinesterase inhibitor, prednisolone, and tacrolimus which was then switched to rituximab administration. Allergy occurred on the second course of rituximab infusion, followed by a myasthenic crisis with rapid worsening and severe weakness requiring intubation and intensive care. At enrollment, she presented with moderately severe weakness in limbs and breath, scoring 12 in QMG score, despite receiving continuous pyridostigmine (360 mg/d), prednisolone (30 mg/d), and regular intravenous immunoglobulin (IVIg, 2.0 g/kg, every 4 weeks) (Fig. 1A).

Approximately $6.161 \times 10^7$ CAR-T cells ($1.01 \times 10^6$ cells/kg) were infused at April 14, 2022. The CAR-T cells proliferated in a rapid multi-log expansion during the first 2 weeks, with peak expansion occurred at 10 days after infusion, and then gradually declined within 3 months (Fig. 1B,C). Following treatment with CAR-T cells, she experienced transient grade 1 CRS (pyrexia with maximum temperature 39.3 °C) at day 8, which resolved within 1 day automatically. No immune effector cell-associated neurotoxicity syndrome, other neurologic toxic effects, or dose-limiting toxicity were observed. Cytopenia, including neutropenia and lymphocytopenia, were observed within 1 month post infusion, likely due to lymphodepletion therapy prior to transfer and due to CAR T-cell expansion, and all resolved within 4 weeks (Fig. 1D). During regular monitoring, cytomegalovirus (CMV) infection was found at 3-week visit without clinical manifestations and was treated by intravenous ganciclovir. The dynamic changes in the protein level of serum inflammatory mediators, including interleukin (IL)-2R, IL-4, IL-5, IL-6, IL-10, tumor necrosis factor (TNF)-α, interferon (IFN)-γ, ferritin, C-reactive protein, and procalcitonin were also recorded, with peak cytokine release occurring within the first 2 weeks post infusion but no persisting beyond 12 weeks (Fig. 1E).

By 3-month post infusion, grip strength, and vital capacity reached approximately normal levels. Functional disability measurement scores, including MG-ADL, QMG, MG-QOL, and mRS demonstrated significant reduction from baseline by 1 year, with maintenance at cutoff date (Fig. 2A). These signs of clinical remission were paralleled by serologic remission with the rapid decrease of anti-AChR and anti-Titin antibodies (Fig. 2B). Pyridostigmine was still given, but tapered to 60 mg/d, without any other concomitant treatment.

**MG-2** was a 60-year-old woman with MuSK-IgG4 seropositive, predominant involvement in bulbar muscles, showing pharyngeal, and tongue weakness. She had a 20-year-long history of MG, and received regular pyridostigmine (180–270 mg/d), prednisone (15 mg/d), and tacrolimus (2.5–3 mg/d), which was later switched to regular rituximab. She failed to respond adequately to these therapies, and impending myasthenic crisis occurred once or twice per year before enrollment, with sequelae of difficulties in chewing and swallowing, limited walking ability, poor scores (QMG 18, MG-ADL 11, QOL 25) on the myasthenia gravis strength and function scales (Figs. 1A and 2A).

Approximately $5.040 \times 10^7$ CAR-T cells ($0.96 \times 10^6$ cell/kg) were infused at May 5, 2022. The CAR-T cells proliferated rapidly and peaked at 10 days after infusion, and then gradually declined within 6 months (Fig. 1B,C). No CRS, immune effector cell-associated neurotoxicity syndrome, other neurologic toxic effects, or dose-limiting toxicity were observed after CAR-T-cell infusion. Levels of serum inflammatory mediators were shown in heatmap, with peak occurred within 2 weeks post infusion. Cytopenia (neutropenia and lymphocytopenia) of grade 3 or higher was also observed following lymphodepletion and CAR-T therapy, and all resolved within 4 weeks (Fig. 1D). CMV infection was found by regular monitoring at a 4-week visit without clinical manifestations, and was treated with oral ganciclovir. One treatment-related severe AE requiring hospitalization (pneumonia) occurred at 10-week, accompanied by subsequent D-dimer increase, and was resolved by intravenous antibiotics.

The patient showed improvement in vital capacity, was able to speak clearly, and take in food easily by 3 months post treatment. By 6 months after treatment, the patient markedly improved in physical function, achieving up to 11-point reduction in MGADL and 16-point reduction in QMG (Fig. 2A). Serum MuSK-IgG was gradually decreased, and maintained negative beyond 1 year (Fig. 2B). Pyridostigmine was still given, but tapered to 90 mg/d, without any other concomitant treatment.

## Immune alterations following CAR-T therapy

Circulating B cells were undetectable by flow cytometry within the first 2 months in both patients, while Patient MG-2 remained in a state of B-cell aplasia beyond 9 months (Fig. 2C,D). B lineage cells gradually returned to the normal ranges at 18 months in both patients, Notably, ~80% of the reconstituted B cells at 18 months were naive B cells, with non-switched, switched memory B cells, and plasma cells being downtrend (Fig. 2D,E). There was a significant decrease in total immunoglobulin after CAR T-cell infusion in both patients, with a slight increase observed in Patient MG-1 (Fig. 2F). Despite these changes, regular IVIg replacement was not necessary regardless of serum Ig levels. These findings suggest a reconstitution of B-cell lineages with more naive phenotype and prolonged suppression of humoral immune response after CAR-T therapy.

We next sought to identify the possible mechanisms underlying the therapeutic efficacy of CAR-T cells in MG. To that end, we conducted longitudinal single-cell RNA sequencing of peripheral blood mononuclear cells from the patients at baseline, at 1 month, and at 3 months post infusion. We integrated these data into a joint dataset of 45,308 single-cell transcriptomes (14,703 cells from MG patients at baseline, 19,729 cells at 1 month post treatment, 10,876 cells at 3-month post treatment, average $22,654 \pm 3617$ cells per donor, Figs. 3A and EV1A,B). As expected, strong differences in cell-type composition were observed between patients pre- and post infusion, with B cells significantly reduced at 1 month and partially restored 3 months post infusion, while myeloid cells decreased progressively (Fig. 3A). We then defined a signature score of inflammation for each cell based by the expression of the reported inflammatory response genes (Liberzon et al, 2015; Ren et al, 2021a), and used the scores to assess the level of contribution to overall inflammation for each cell. Interestingly, we found high inflammatory score expression in the clusters of myeloid cells,

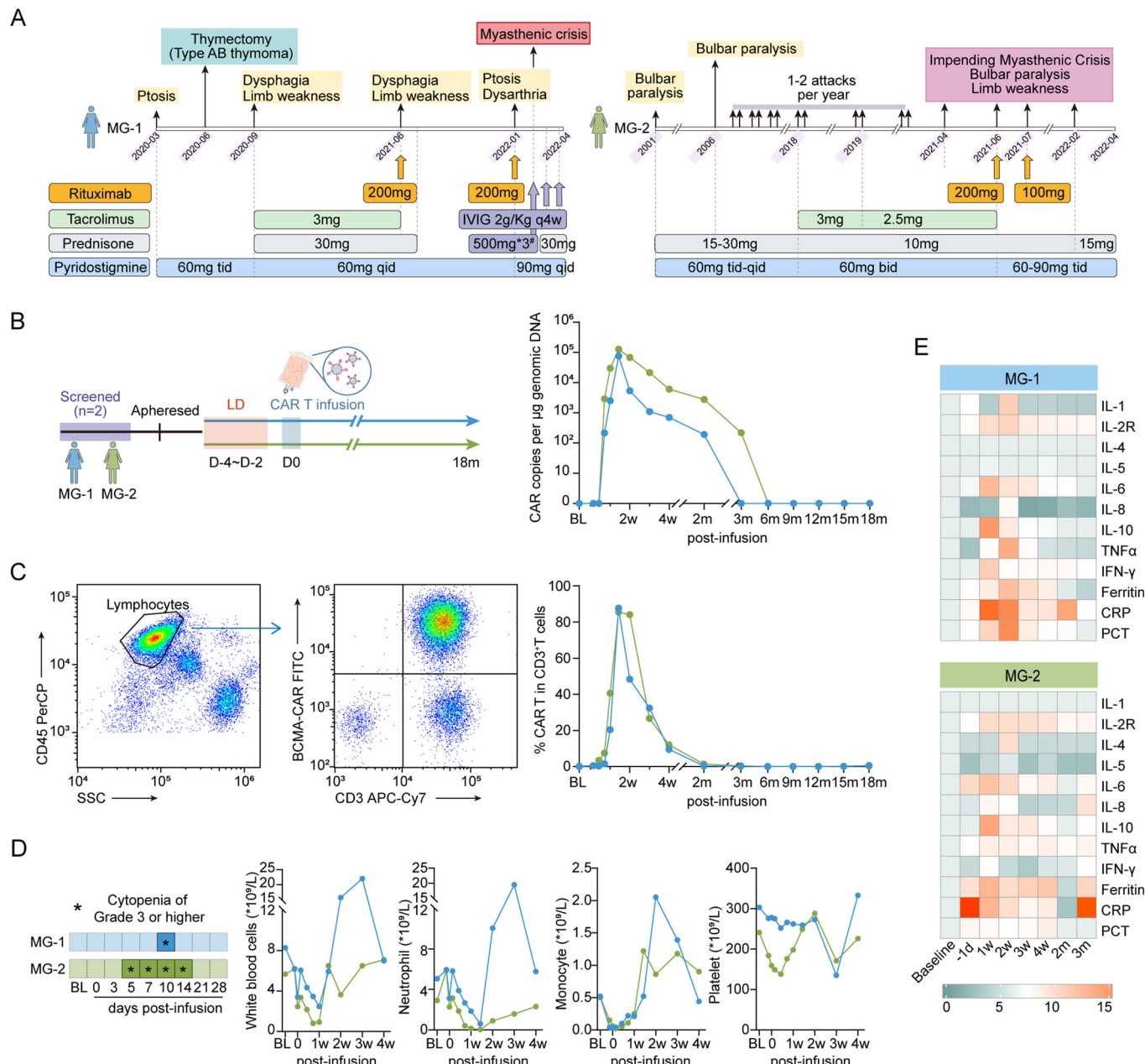

**Figure 1. CAR T-cell kinetics and inflammatory mediators release following infusion.**

(A) A schematic overview of the time points at which patients receive different treatments before CAR-T infusion. #Patient MG-1 was treated with IVIG 2 g/Kg + intravenous pulse steroid 500 mg* 3days for myasthenia crisis. (B) A schematic overview of CAR-T treatment procedure. CAR T-cell kinetics are shown by the CAR copies per µg genomic DNA at serial time points post infusion detected by droplet digital PCR. (C) Representative plots showing FACS analysis stained for CAR-T cells with FITC-labeled human BCMA Fc tag protein and APC/Cy7 anti-human CD3 antibody in patient MG-1 at day 10 after CAR T-cell infusion. CAR T-cell percentage in circulating CD3+ T lymphocytes at serial time points after treatment. (D) Timelines of patients with cytopenia of grade 3 or higher at baseline and indicated time points after CAR T-cell infusion. BL baseline. Kinetic changes in numbers of circulating total white blood cells, neutrophils, monocytes and platelets. (E) Heatmap depicting protein levels of inflammatory mediators in blood following CAR T-cell infusion. Interleukin IL, TNF tumor necrosis factor, IFN interferon, CRP C-reactive protein, PCT procalcitonin. Average levels are normalized from the baseline. Source data are available online for this figure.

which progressively downregulated at 1 month and 3 months post infusion (Fig. 3A).

We then performed paired transcriptional profiling and B-cell repertoire sequencing at the single-cell level (Fig. 3B). A substantially predominant clonal expansion of B cells, particularly plasmablasts and plasma cells (PB/PCs), was observed in patients at

baseline, suggesting abnormally expansion of PB/PCs in these patients. Moreover, we identified 1272 differentially expressed genes when comparing B cells at baseline with reconstituted B cells at 3-month post treatment (Bonferroni-adjusted $P$ value <0.05). The downregulated genes were enriched for signatures associated with Ig production, activation of immune response, adaptive

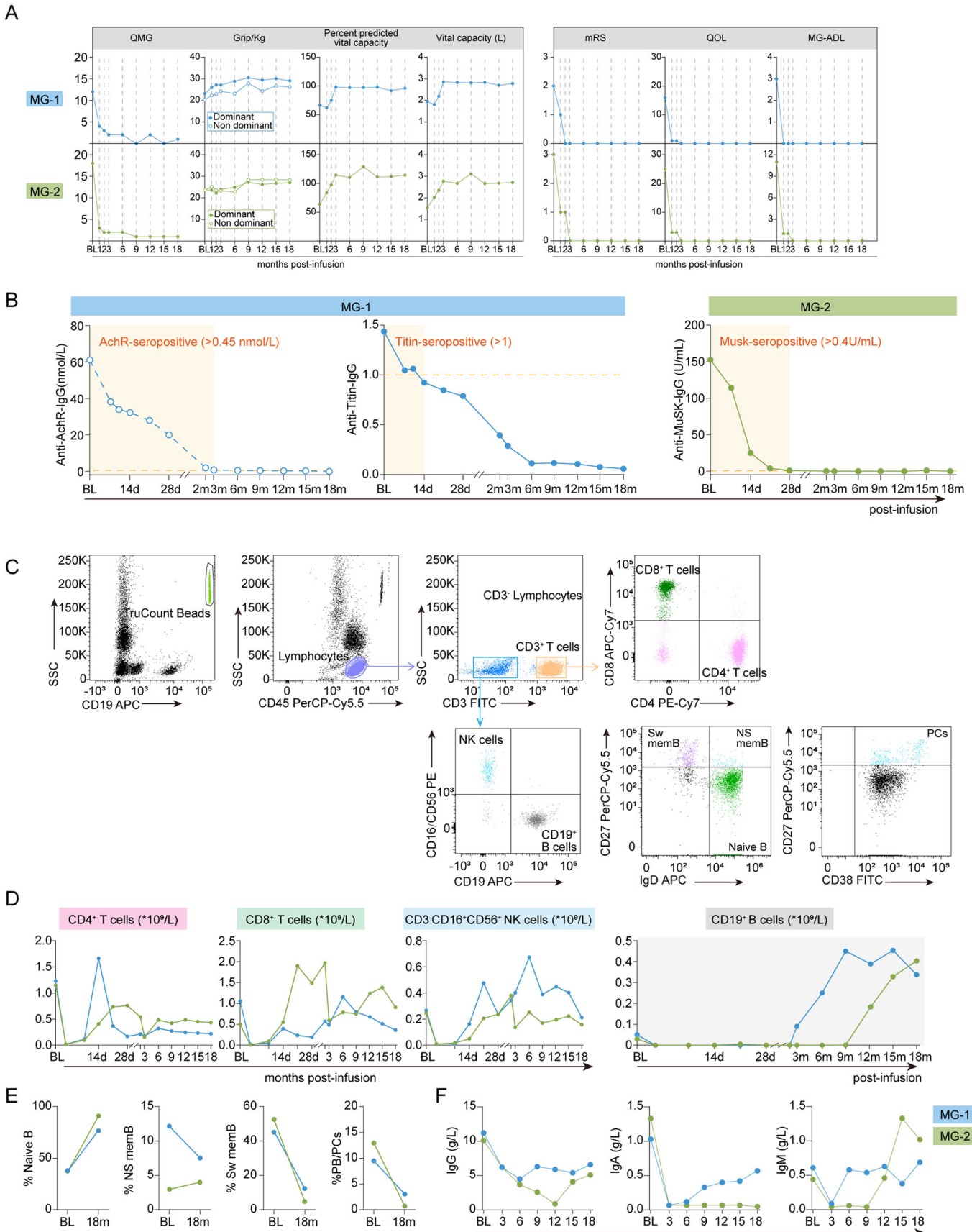

**Figure 2. Clinical evaluation following CAR T-cell infusion.**

(A) Kinetic parameters of Patient MG-1 and MG-2, including QMG score, grip strength, vital capacity volume and percent predicted vital capacity, mRS score, the MG-QOL15 questionnaire, and MG-ADL scale score. (B) Pathogenic antibodies levels in serum (anti-AChR-IgG and anti-Titin-IgG for MG-1, anti-MuSK-IgG for MG-2). (C) Representative images showing the gating strategies for lymphocyte subset analysis using TruCOUNT beads, and B-cell lineages stained for CD19$^+$ CD27$^-$ IgD$^+$ naive B cells, CD19$^+$ CD27$^+$ IgD$^+$ non-switched memory B cells (NS mem), and CD19$^+$ CD27$^+$ IgD$^-$ switched memory B cells (Sw mem) and CD19$^+$CD27$^+$CD38$^{high}$ plasma cells (PCs). (D) Kinetic changes in numbers of CD4$^+$ T cells, CD8$^+$ T cells, CD3$^-$CD16$^+$CD56$^+$ NK cells, and CD19$^+$ B cells at baseline and at indicated time points after CAR T-cell infusion. (E) Changes in the percentage of naive B cells, non-switched memory B cells, switched memory B cells and plasma cells between baseline and 18-month post infusion. (F) Changes in total immunoglobulin levels (IgG, IgA, and IgM) before CAR T-cell therapy and at 18-month post infusion. Source data are available online for this figure.

immune response, and B-cell-mediated immunity (Fig. 3C). This suggests that CAR-T therapy led to the reconstitution of B-cell lineages with more naive phenotype and suppressed humoral immune response.

To further investigate the impact of CAR-T cells on the immune system, we conducted a single-cell analysis to examine immune cell interactions. Cell-cell interaction prediction using Cellchat (Jin et al, 2021), revealed a higher frequency of crosstalk in patients at baseline (Figs. 3D and EV2A,B). A remarkable upregulation of inflammatory cytokines, especially macrophage migration inhibitory factor (MIF), was observed in B-cell subtypes from the patients, whose corresponding receptors (CD74/CXCR4, CD74/CD77) were also upregulated in other immune cells, especially myeloid subclusters, indicating that the abnormally expanded B cells might exert distinct roles in activating myeloid cells via MIF secretion. Given that MIF is a key driver of immunopathogenesis in rheumatic disorders (Kang and Bucala, 2019), and soluble BCMA (sBCMA) is a widely used surrogate marker for PB/PCs that is detectable in circulation (Mikkilineni and Kochenderfer, 2021), we measured the serum levels of MIF and sBCMA in an independent cohort of MG ($n = 47$), and age- and sex-matched controls ($n = 47$) using ELISA (Fig. 3E). The results further confirmed the higher expression of MIF in the serum of MG patients and sBCMA levels positively correlated with MIF expression (Fig. 3E), further suggesting the association between PB/PCs expansion and MIF secretion in MG. Interestingly, the serum levels of MIF and sBCMA correspondingly decreased at 3 months post infusion (Fig. 3F). Meanwhile, GSEA analysis results showed that pathways related to immune responses and inflammation were largely downregulated in endogenous T cells, NK cells and myeloid cells progressively post treatment (Fig. 3G). Thus, our results raise the possibility that CAR-BCMA T cells may not only direct suppress humoral immune responses of B cells but also reduce systemic inflammation by eliminating abnormal interaction between PB/PCs and other circulating immune cells.

## CAR-T-cell characteristics after infusion in patients with MG

We next aimed to understand how the host immunological milieu shaped the transferred CAR-T cells in MG. We performed a comparative single-cell transcriptional and TCR analysis of the CAR-T cells in IPs before transfer using a Cellular Indexing of Transcriptomes and Epitopes by Sequencing (CITE-seq) approach, and the CAR-T cells in vivo at 1 month after infusion sorted by flow cytometry (Fig. EV3A). In total, we identified 7249 CAR-T

cells in IPs and 2695 CAR-T cells in vivo with high quality. Re-clustering of endogenous T cells and CAR-T cells identified distinct cell subclusters into naive (Tn, CCR7$^+$ LEF1$^+$), central memory (Tcm, CCR7$^+$ GPR183$^+$), effector memory (Tem, CCR7$^-$ GPR183$^+$ GZMK$^+$), CD8$^+$ effector (Te, CD8A$^+$ GZMB$^+$ NKG7$^+$), CD4$^+$ cytotoxic cells (CD4$^+$ CTLs, NKG7$^+$), and CD4$^+$ regulatory T cells (FOXP3$^+$) using canonical markers ("Methods", Fig. EV3B,C). During the manufacturing process, IP cells are stimulated which may affect the expression of genes related to proliferation (MKI67, STMN1) and markers of cytotoxicity (GZMB, NKG7). This modulation could potentially impact the classification of cells. Transcriptomic analysis revealed strong evidence of ongoing proliferation in CAR-T cells within IPs, as evidenced by the elevated expression of PCNA, TOP2A, and MKI67, as well as cell cycle scoring using gene sets to identify cells in S phase or G2/M phase (Melenhorst et al, 2022) (Fig. EV3B–D). In IPs, ~58.7% of CAR-T cells in IPs were found to be in S, G2 or M phase, compared with only 15.6% of endogenous T cells at baseline and 17.2% of CAR-T cells at 1 month post infusion in vivo (Fig. EV3D). We further distinguished the cycling effector phenotype (cycling Te, MKI67$^+$ GZMB$^{lo}$), with a balanced mix of CD4$^+$ and CD8$^+$ cells, most prevalent in CAR-T cells in IPs. We also identified several cells as Te, Tcm and Tem, and few cells as Tn in IPs (Fig. 4A,B). At 1 month, CAR-T cells in vivo were composed primarily of CD8$^+$ Te, CD8$^+$ Tem, and CD4$^+$ Tem cells (Fig. 4A,B). Despite changes in cellular components after transfer, both CD4$^+$ and CD8$^+$ CAR-T cells at 1 month in vivo exerted a functional phenotype with lower proliferation and energy metabolism and higher cytotoxicity compared with CAR-T cells in IPs (Fig. 4C).

We next used our TCR enrichment libraries to monitor the changes in CAR-T phenotypes over time. The dominant CD8$^+$ TCR clones at 1 month in vivo originated primarily from endogenous Te and Tem cells at baseline, manufactured into MKI67$^+$ GZMB$^{lo}$ cycling effector Te cells in IPs and differentiated into Te at 1 month after transfer (Fig. 5A). Notably, KLRF1$^+$ GZMB$^{hi}$ CD8$^+$ Te2 subset was found to be increased in frequency at 1 month, with elevated expression of genes related to NK receptor (KLRG1, KLRF1) (Fig. EV3B), gene signature implicated previously in T-cell dysregulation (Good et al, 2021). In contrast, CD4$^+$ CAR-T cells demonstrated distinct dynamics, where top CD4$^+$ TCR clones in vivo originated diversely. CD4$^+$ CAR-T cells in the IPs exhibited fewer cells in the cell cycle compared to CD8$^+$ subsets, providing a possible explanation for their lower proliferation and less frequency after transfer.

We next compared the "D28-clones" of CAR-T cells in IPs that could be detected at 1 month in vivo with other CAR-T clones

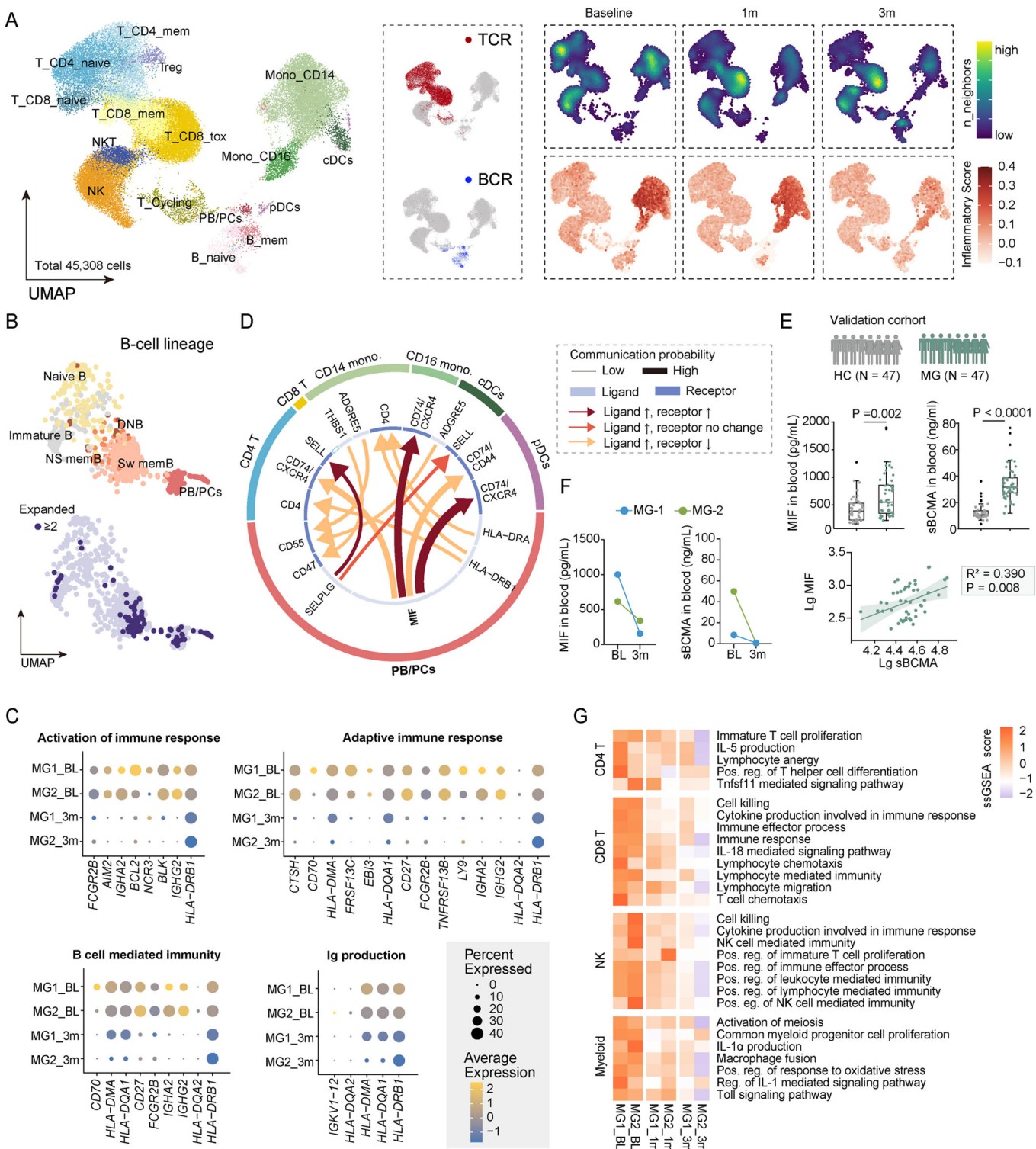

in IPs that disappeared at 1 month after transfer (Fig. 5B). A significantly higher fraction of CD8$^+$ cycling CAR$^+$ subclusters in IPs and CD8$^+$ Te cells at baseline were observed in these "D28-clones" (Fig. 5B), suggesting proliferating cytotoxic-like CD8$^+$ CAR-T subsets mainly manufactured from Te cells before treatment play a more predominant role in MG.

## Distinctive features of CAR-T cells from myasthenia gravis

Finally, we explored the distinct molecular features of endogenous T cells and CAR-T cells from the MG patients. To that end, we integrated our dataset of endogenous T cells at baseline and CAR-

**Figure 3. Immune alterations following CAR T-cell therapy.**

(A) Uniform manifold approximation and projection (UMAP) plot of 45,308 single-cell transcriptomes of peripheral blood mononuclear cells integrated from the two patients at baseline, at 1 month and at 3 months post infusion. Clusters denoted by color are labeled with inferred cell types, including three CD4⁺ T-cell clusters, three CD8⁺ T-cell clusters, cycling T cells (T_Cycling), NK cells, NKT cells, two monocyte clusters, conventional dendritic cells (cDCs), plasmacytoid dendritic cells (pDCs), and three B-cell clusters. UMAP of cells colored by BCR and TCR detection. Feature plots colored by cell density (top) and a signature score of inflammation for each cell based on the expression of inflammatory response genes (bottom). (B) UMAP plots showing re-clustering of B cells colored by six subsets, annotated as immature, naive, non-switched memory (NS mem), switched memory (Sw mem), double negative (DN) B cells, and plasmablasts and plasma cells (PB/PCs), and colored by clone size. (C) Mean expression of differentially expressed genes between baseline and 3-month post infusion, enriched in activation of immune response, B-cell-mediated immunity, and Ig production. (D) Circos plot showing the ligand-upregulated interactions mediated by ligand-receptor pairs between PB/PCs and other immune cells. The outer ring displays color-coded cell types, and the inner ring represents the involved ligand-receptor interacting pairs. The line width is proportional to the communication probability in MG comparing to control group. Colors and types of lines are used to indicate different types of interactions as shown in the box. DC dendritic cells, Mono monocytes. (E) A schematic overview of the validation cohort of MG patients. HC, N = 47; MG, N = 47. Boxes denote the interquartile range (IQR), and the median is shown as horizontal bars. Whiskers extend to 1.5 times the IQR, and outliers are shown as individual black dots. Group comparisons were computed with a two-sided Wilcoxon rank-sum test with a Benjamini–Hochberg correction. Scatter plots depicting the correlations between sBCMA (lg) and MIF (lg). Biomarker values were log10-transformed to reduce skewness. The solid lines indicate the regression line and the 95% confidence intervals. The correlation coefficients and P values from the partial correlation analysis are shown. (F) Scatter plots depicting serum levels of sBCMA and MIF quantified by ELISA in the patients at baseline and 3 months post infusion. (G) Heatmap showing single-sample GSEA scores of indicated signatures in CD4⁺ T cells, CD8⁺ T cells, NK cells, and myeloid cells of patients at baseline and at 3-month post infusion. Source data are available online for this figure.

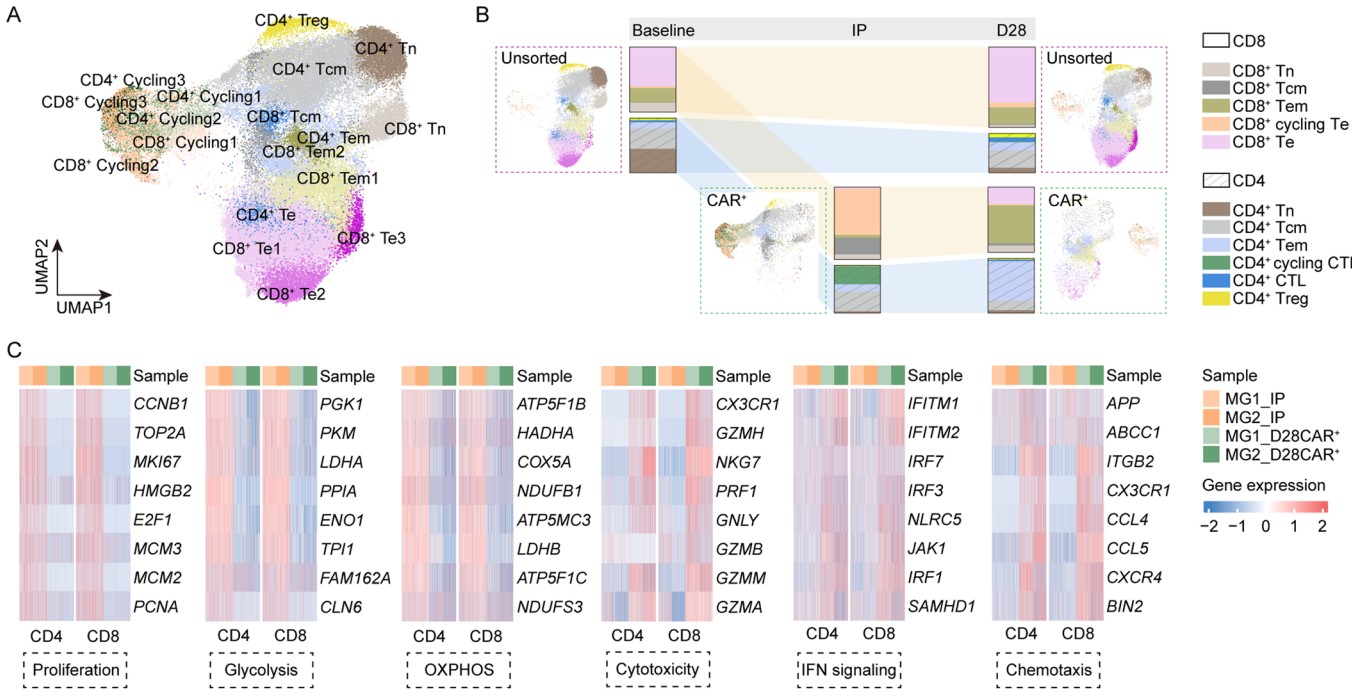

**Figure 4. Transcriptional signature of CAR-T cells in MG.**

(A) UMAP plot of CAR-T cells in IPs and at 1 month post infusion, and endogenous T cells at baseline and at 1 month post infusion. (B) Depiction of T-cell subset frequencies at each timepoint. Bar widths are proportional to the fraction of cells being classified as a particular subset. (C) Expression of differentially expressed genes indicating lower proliferation, glycolysis, OXPHOS, and higher cytotoxicity, IFN signaling, chemotaxis of CAR-T cells collected at 1 month compared with CAR-T cells in IP. Source data are available online for this figure.

BCMA T cells in IPs from the 2 patients, with three published external datasets (GSE197851, GSE151310, GSE197268) (Haradhvala et al, 2022; Li et al, 2021; Rodriguez-Marquez et al, 2022), including (1) CAR-BCMA T cells in IPs from 3 healthy donors, (2) CAR-BCMA T cells in IPs from 1 patient with plasma cell leukemia, (3) T cells in 12 individual blood samples at baseline and CAR-CD19 T cells in 18 IPs from lymphoma patients treated with Axi-cel, and T cells in 8 blood samples and CAR-CD19 T cells in 13 IPs from lymphoma patients treated with Tisa-cel (Fig. 6A). We identified diverse T-cell subtypes present based on canonical

markers. The analysis also showed that the cell clusters were interspersed across different datasets, indicating less variability among patients or cohorts than among types of cells (Fig. 6A). We observed distinct clustering of the MKI67⁺ cycling CAR⁺ T cells in all IP samples, suggesting potential CAR-driven differences in their phenotype that shared across cohorts and products (Fig. 6A).

We next analyzed transcriptional differences among the CAR⁺ CD8⁺ cycling Te cells in our IPs with those in other products separately (Fig. 6B). Interestingly, we detected reduced expression of genes encoding cytotoxic signatures (GZMA, GZMB, GNLY), cell cycle

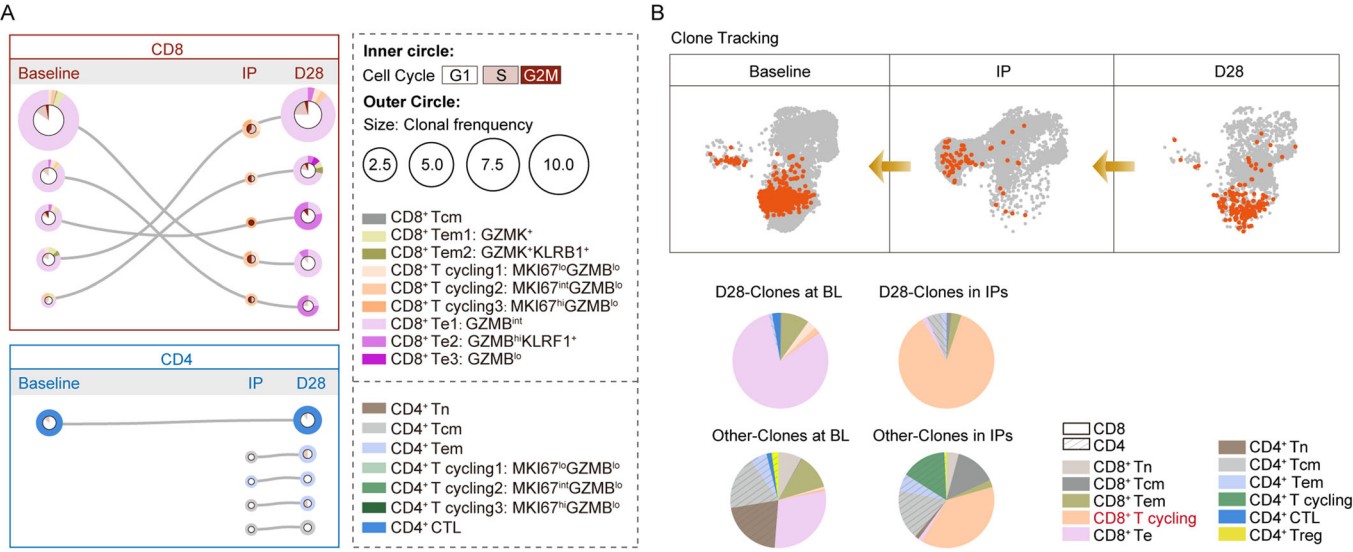

**Figure 5. Clone tracking of CAR-T cells in MG.**

(A) The top 5 most prevalent TCR clones identified at 1 month post treatment and the corresponding clones in IPs or at baseline are shown for CD8⁺ and CD4⁺ CAR-T subsets. For each, circles show the clone belongs at each timepoint, with sizes corresponding to the clone frequency in its sample. Pie charts of the inner circle showing the distribution of cells in each phase of the cell cycle. Pie charts of the outer circle showing the distribution of cells in each subset. (B) UMAP plots and pie charts showing the distribution of cells in patients at baseline and CAR-T cells in IPs that finally exist in vivo at 1 month post infusion, by clone tracking. Source data are available online for this figure.

molecule *CDKN1A* and energy metabolism regulators (*ATP5F1D, COX5A, NDUFA3*), but elevated levels of activation/exhaustion marker *FYN*, checkpoint regulator *CBLB* and genes related to mitochondrial function (*PSENEN, PRKACB, HSF1*) (Fig. EV4A), perhaps suggestive of an early dysfunction potential of our product. We then applied Ingenuity Pathway Analysis (IPA) to link these differentially expressed genes (DEGs) expression, with literature-based canonical pathways. Consistently, IPA results disclosed that the functional programs in MG were downregulated cell cycle, suppressed oxidative phosphorylation and glycolysis, as well as profound mitochondrial dysfunction (Fig. 6B), verifying the possible functional and metabolic features underlying the characteristics of CAR-T cells in MG.

Similar analyses were implemented comparing basal CD8⁺ Te cells from MG patients with the cells from lymphoma patients. As expected, we also observed lower expression of cytotoxic, cell cycle and energy metabolism signatures, upregulation of genes related to mitochondrial function, NK receptor (*KLRC2, KLRC3*) and *FYN*, inhibited cell cycle and metabolism pattern and activated mitochondrial dysfunction in Te cells from the MG patients at baseline, compared with Te cells from lymphoma patients in external cohorts (Figs. 6C and EV4B). These results indicated that the compromised cytotoxicity and proliferation property, and metabolic signature of the CAR-T cells in MG might result from the dysregulated function of basal Te cells from autoimmunity.

## Discussion

Our preliminary data suggest that CAR-BCMA T cells are well-tolerated and highly effective in treating refractory MG, demonstrated by the improvement in physical function and serologic remission. Patient MG-1 with anti-AChR-IgG and anti-Titin antibodies, and patient MG-2 with anti-MuSK-IgG4 represent the two main subtypes of MG, respectively. Both patients showed poor response to first-line immunosuppressive therapies. The B-cell-targeting therapy rituximab was considered as an early therapeutic option but also failed in these two patients. In regard to these refractory generalized MG patients, CAR-T therapy was discussed and then initiated with written consent. Notably, minimal symptom expression was achieved following one dose CAR-T infusion, which was sustained despite the cessation of all the immunosuppressive therapy.

Although increasing innovative strategies for the treatment of MG were officially approved or showed promising efficacy in various clinical trials (Mane-Damas et al, 2022), yet there are still a proportion of patients remained highly recurrent and refractory. Several recent case reports have raised the possibility of CAR T-cell therapy in treating patients with refractory MG. One clinical trial showed CAR-T cells engineered with RNA targeting BCMA without lymphodepletion have therapeutic potentials in treating 9 MG patients during a 5-months observation period, but this RNA CAR-T treatment required weekly infusions for consecutive 6 weeks (Granit et al, 2023). Another recent report also observed clinical efficacy and primary safety of conventional DNA-engineered anti-CD19 CAR-T treatment in MG, although the follow-up persisted only 62 days (Haghikia et al, 2023). In this report, our DNA-engineered anti-BCMA CAR-T cells, although required preconditioning chemotherapy which could allow favorable condition for CAR-T-cell proliferation (Geyer et al, 2019; Hirayama et al, 2019; Narayan et al, 2022; Turtle et al, 2016), showed prolonged efficacy following only one single dose. Furthermore, it is worth noting that all the previous reports only offered short-term observation with median follow-up no more than 8 months while the present case

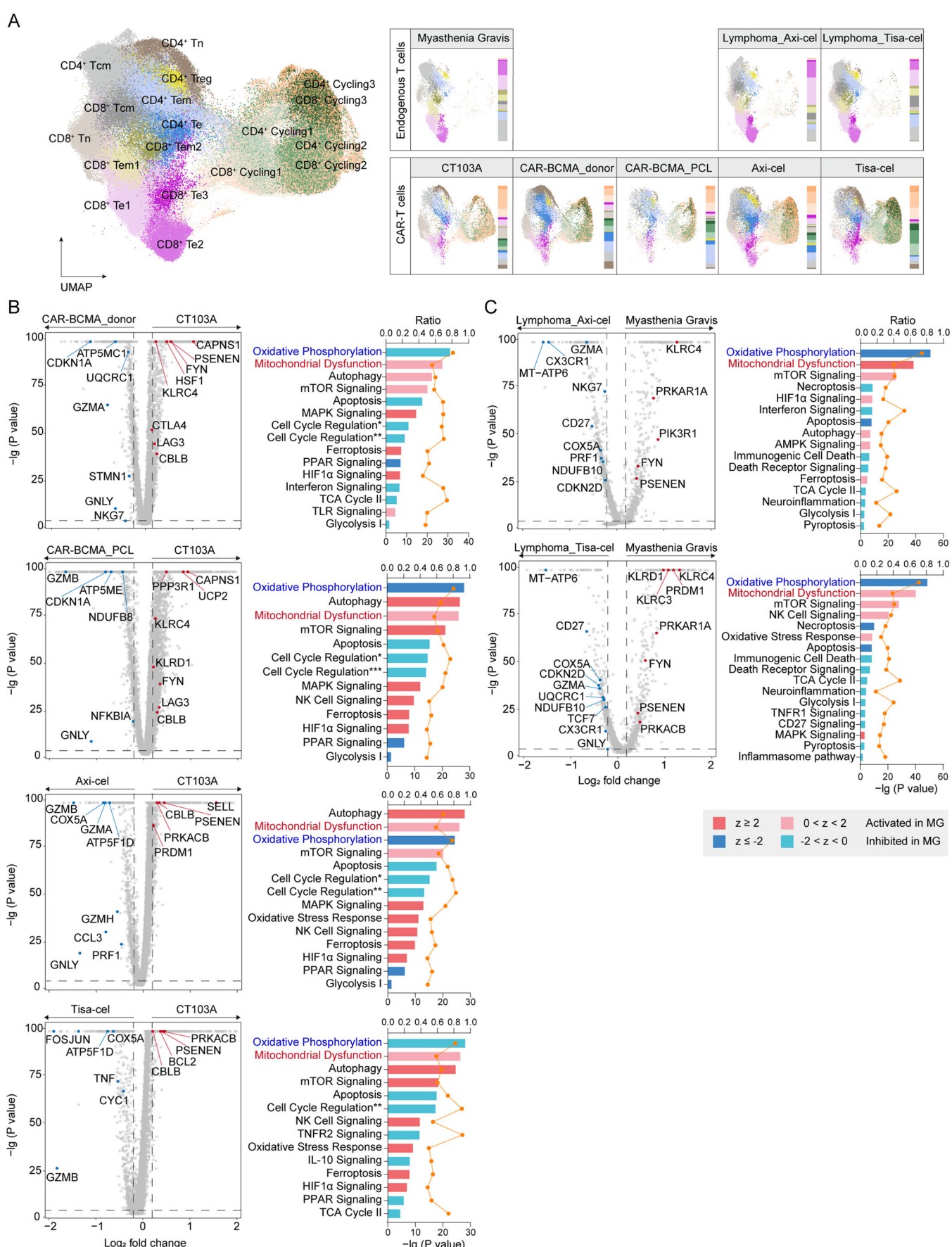

**Figure 6.  Distinct signatures of CAR-T cells from patients with MG.**

(A) UMAP plots showing the integrating endogenous T cells and CAR-T cells colored by subclusters. Single-cell transcriptomics in three recently published external datasets (GSE197851, GSE151310, GSE197268), including (1) CAR-BCMA T cells in IPs from three healthy donors, (2) CAR-BCMA T cells in IPs from 1 patient with plasma cell leukemia, (3) T cells in 12 individual blood samples at baseline and CAR-CD19 T cells in 18 IPs from lymphoma patients treated with Axi-cel, and T cells in 8 blood samples and CAR-CD19 T cells in 13 IPs from lymphoma patients treated with Tisa-cel, along with our dataset including T cells at baseline and CAR- T cells in IPs (CT103A) from the two patients with MG, were used for signature validation. (B) Comparison of differentially expressed genes between CAR$^+$ CD8$^+$ cycling Te cells in the IPs from patients with MG and other IPs. CT103A, $N = 4044$ cells; CAR-BCMA_donor, $N = 4739$ cells; CAR-BCMA_PCL, $N = 2111$ cells; Axi-cel, $N = 11,392$ cells; Tisa-cel, $N = 16,348$ cells. Group comparisons were computed with a Wilcoxon rank-sum test with a Bonferroni correction. Ingenuity Pathway Analysis (IPA) was performed showing corresponding signaling pathways regulated by these DEGs. z score reflects the predicted activation level (z ≥ 2 or z ≤−2 can be considered significant). The yellow curve denotes the ratio between the number of the DEGs and the total number of genes in each of these pathways. Cell Cycle Regulation*, Cell Cycle: G1/S Checkpoint Regulation; Cell Cycle Regulation**, Cell Cycle: G2/M DNA Damage Checkpoint Regulation; Cell Cycle Regulation***, Cyclins and Cell Cycle Regulation. (C) Volcano plot showing comparison of differentially expressed genes between endogenous Te cells from the MG patients at baseline and Te cells from other groups. Myasthenia Gravis, $N = 2898$ cells; Lymphoma_Axi-cel, $N = 7123$ cells; Lymphoma_Tisa-cel, $N = 4857$ cells. Group comparisons were computed with a Wilcoxon rank-sum test with a Bonferroni correction. Ingenuity Pathway Analysis (IPA) was performed showing corresponding signaling pathways regulated by these DEGs. z score reflects the predicted activation level (z ≥ 2 or z ≤−2 can be considered significant). The yellow curve denotes the ratio between the number of the DEGs and the total number of genes in each of these pathways. Source data are available online for this figure.

series illustrate great potentials of CAR-T-cell therapy in treating seropositive MG, as evidenced by sustained depletion of auto-antibodies and prolonged therapeutic efficacy for up to 18 months.

The safety concerns associated with CAR-T treatment may be a dilemma for its application in autoimmune diseases. According to the available clinical trials of CAR-T therapy on autoimmune diseases and hematological malignancies, most adverse effects occurred in the early stage. To date, no prolonged or delayed adverse events have been reported in the treatment of autoimmune diseases during the longest follow-up of 17 months (Mackensen et al, 2022; Muller et al, 2023; Qin et al, 2023). While in hematological tumors, several cases of prolonged toxicity were reported. Two patients with multiple myeloma had developed a grade 3 cerebellar disorder that was considered to be a dose-limiting-toxicity effect for the dose of 450×10⁶ CAR-T-anti-GPRC5D at 6.5 and 8.4 months, respectively (Mailankody et al, 2022). In another clinical trial using anti-BCMA CAR-T cells, the patient developed movement disorder with features of parkinson-ism at day 101 after CAR-T-cell infusion (Van Oekelen et al, 2021). These neurological adverse events were both speculated to be related with target antigens expressing in the brain tissue. In addition, prolonged grade 3/4 cytopenia such as neutropenia, thrombocytopenia, anemia and lymphocytopenia, were also reported in several patients, with the susceptibility to develop fatal infectious and hemorrhagic events (Benjamin et al, 2020; Mei et al, 2021; Munshi et al, 2021). In our study, we also pay a lot of attention on side effects of CAR-T therapy, including both short-term CRS and possible prolonged toxicity. Fortunately, in our two cases, short-term AEs resolved within early stage. No immune effector cell-associated neurotoxicity syndrome, other neurologic toxic effects, or dose-limiting toxicity were observed. With regular follow-up performed every 3 months for more than 1.5 years, no delayed adverse effects were observed. Nonetheless, subsequent regular follow-up and continuous monitoring are still needed for the prevention of any potential delayed or long-term side effects. In addition, more specific patient-tailored strategies, which would minimize the considerable risk of side effects are warranted for treating patients with MG.

Interestingly, our study proved the notion that the highly clonally expanded PB/PCs in MG were generally not influenced by regular/multiple immunomodulatory therapies, consistent with the findings in those highly refractory patients with MG (Jiang et al, 2020), peripheral neuropathy (Maurer et al, 2012), and pemphigus vulgaris (Colliou et al, 2013), which might underlie their resistance to previous therapies. Inspiringly, CAR-BCMA T cells demonstrated promising efficacy to eliminate these BCMA-expressed cells, PB/PCs mainly. Given that autoantibodies progressively decreased and remained nearly undetectable for over 18 months, the newly grown precursor B cells might not differentiate into pathogenic PB/PCs after treatment. However, the mechanism underlying the reconstitution of B-cell lineage is currently unknown and needs further exploration. Moreover, our data also raise the possibility that the abnormally expanded PB/PCs might activate other immune cells and exacerbate inflammation via enhanced secretion of cytokines such as MIF. Thus, reconstitution of B-cell lineages-reminiscent of a B lineage-specific immune reset, diminished pathogenic autoantibodies, and normalization of the potential overactivated B-cell-related immune microenvironment might underlie therapeutic efficacy of CAR-BCMA T cells in these patients.

Despite the favorable safety profiles and promising therapeutic potentials, the shorter persistence of CAR-T cells (approximately 30 days) in autoimmune diseases compared to that in hematological cancers, was observed in early reports (Mackensen et al, 2022; Qin et al, 2023) and the present study. Our analysis provides significant insights into the temporal changes in clonal and expression dynamics of CAR-T cells in MG, revealing several common characteristics of CAR T-cell behavior pre- and post infusion that are shared or distinct across hematological malignancies and autoimmune diseases (Li et al, 2021; Van Oekelen et al, 2021). Unlike cancers, however, circulating B cells/plasma cells are easily and rapidly cleared, resulting in limited stimulation of functional effector T cells that persist after targeting cell eradication. In addition, our study highlights the importance of infusing proliferating CAR$^+$ Te cells manufactured from endogenous Te phenotype in the final expansion stage of autoimmunity. Several studies in preclinical models provide evidence that anti-CD19 lymphocytes manufactured from enriched Te or Tem cells are less effective in target clearance compared to CAR-T cells derived from central memory or naive T cells (Gupta and Gill, 2021; Sommermeyer et al, 2016). This limitation may also hinder the effectiveness of CAR-T cells in treating autoimmunity. Furthermore, both experimental and clinical findings indicate that optimal CAR-T-cell-mediated function requires efficient cytotoxicity and

sufficient persistence for the successful treatment of hematological malignancies and solid tumors (Katsarou et al, 2021). We documented a suppressed effector signature and profound mitochondrial dysfunction of Te cells in patients with refractory MG, which might partially result from multiple/long-term immunosuppressants and steroid use prior to enrollment similar to that observed in systemic lupus erythematosus (Takeshima et al, 2022). The subsequent compromised properties of the CAR-T cells manufactured from these Te cells, provide a possible explanation for their poor persistence in autoimmune diseases. Additionally, inhibited proliferating properties and enhanced cell exhaustion/dysfunction were also observed in basal cells and manufactured CAR-T cells of MG patients. These features might also lead to their relatively shorter persistence and poorer effectiveness, in addition to the limited antigen exposure in autoimmunity. Further investigation in larger cohorts and exploration of the molecular features of CAR-T cells are also warranted to seek more evidence in treating patients with MG.

There are still several challenges inherent to this study. First, we only reported here information on two representative patients with MG (one with AChR-IgG, and one with MuSk-IgG) before and after receiving CAR-T treatment, and a larger cohort is needed for enhancing the reliability of the results. Second, we couldn't fully rule out the possible short-term benefits from previous treatments of steroids and immunosuppressants or preconditioning lymphodepletion, yet the sustained remission over 18 months seems to have slight effects from these treatments.

Exploring the molecular characteristics of CAR-T cells in autoimmunity could optimize their design and manufacturing process, and may also aid in identifying which cell clusters are more effective in treating autoimmune diseases. Further investigation in animal models is warranted to understand how the host immunological milieu influences the behavior of CAR-T cells in vivo, and seek higher efficiency and longer persistence in treating autoimmune diseases.

# Methods

## Human subjects

This study was approved by the Institutional Review Board of Tongji Hospital (TJ-IRB20220101) and conformed to the WMA Declaration of Helsinki and to the principles set out in the Department of Health and Human Services Belmont Report. Two patients with relapsed and refractory MG were recruited at Tongji Hospital, Tongji Medical College, Huazhong University of Science and Technology, China, and written informed consent were obtained. A diagnosis of definite MG with relapsed and refractory disease course as defined below:

(i) Confirmed generalized MG and class II-IV disease according to the Myasthenia Gravis Foundation of America (MGFA) classification system (Narayanaswami et al, 2021); (ii) a positive serological test for antibodies against AChR, MuSK, or lipoprotein receptor-related protein 4; (iii) impaired activities of daily living defined as a Myasthenia Gravis-Activities of Daily Living (MG-ADL) score of 6 or higher; (iv) patients had to have received treatment with one or more immunosuppressive therapies, or with regular intravenous immunoglobulin or plasma exchange for 1 year

without symptom control or side effects that limit functioning (Howard et al, 2017; Sanders et al, 2016).

## Treatment procedure

Anti-BCMA CAR-T cells (CT103A) were successfully manufactured by Nanjing IASO Biotechnology Co., Ltd for the two patients from their autologous T cells, which were transduced with a lentiviral vector containing a fully human anti-BCMA single-chain fragment, and expanded over about 10 days as described in our previous studies (Qin et al, 2023; Wang et al, 2021). Participants stopped all immunosuppressants and steroid one to three days before treatment initiation. Each participant received a three-day consecutive lymphodepletion therapy consisting of cyclophosphamide 500 mg/m$^2$ plus fludarabine 30 mg/m$^2$ ($-4$, $-3$, $-2$ day before infusion), and $1.0 \times 10^6$ total CAR-T cells per kilogram body-weight on day 0 (Fig. 1B). Results were reported here to a data cutoff of December 1, 2023, with follow-up beyond 18 months.

## Safety evaluation

Treatment-related adverse events were documented, and the severity was graded according to the National Cancer Institute Common Terminology Criteria for Adverse Events, version 5.0. Cytokine release syndrome, immune effector cell-associated neurotoxicity syndrome and any other neurotoxicity effects were defined and graded according to published criteria (Lee et al, 2014; Lee et al, 2019).

## CAR vector copy number analysis by droplet digital PCR (ddPCR)

CAR vector copy number was measured by ddPCR conducted following the manufacturer's instructions. In brief, blood samples were collected using Lysis Buffer (BD Biosciences) and genomic DNA was extracted with the DNA Blood Mini Kit (catalog number 51104; Qiagen, Valencia, CA). ddPCR was performed at indicated timepoint in virtue of the probes and primers targeting the scFV sequence as previously reported (Lou et al, 2020; Wang et al, 2021).

## Flow cytometry

Fresh blood samples at indicated time points were collected for flow-cytometry analysis. Circulating numbers of CD4$^+$ T cells, CD8$^+$ T cells, CD3$^-$CD16$^+$CD56$^+$ NK cells, and CD19$^+$ B cells were determined by using TruCOUNT tubes and BD Multitest 6-color TBNK Reagent Kit (BD Biosciences, San Jose, CA, USA) according to the manufacturer's instructions as previously described (Luo et al, 2021). For lymphocyte subset analysis, following antibodies were used: PerCP/Cyanine5.5 anti-human CD45 Antibody (Biolegend, 304028), FITC anti-human CD3 antibody (BD Biosciences, 561802), PE/Cyanine7 anti-Human CD4 (BD Biosciences, 560649), APC/Cyanine7 anti-Human CD8 (BD Biosciences, 557834), APC anti-human CD19 Antibody (Biolegend, 302212), PE anti-human CD16 Antibody (Biolegend, 302056), PE anti-human CD56 Antibody (Biolegend, 318306), FITC anti-Human CD38 (BD Biosciences, 567147), PerCP/Cyanine5.5 anti-Human CD27 (BD Biosciences, 560612). For CAR T-cell percentage analysis, PerCP anti-Human CD45 (BD Biosciences, 347464), APC/Cyanine7 anti-

human CD3 Antibody (Biolegend, 344818) and FITC-labeled human BCMA Fc tag protein (Acrobiosystems, BCA-HF254) were used. All antibodies were used at the manufacturer's recommended concentration. FlowJo v10 was used for analysis.

## Evaluation of clinical activity

Clinical response for MG was assessed with (i) Mean grip strength for both hands separately; (ii) Vital capacity; (iii) the Quantitative Myasthenia Gravis (QMG) score (Barohn et al, 1998); (iv) MG-ADL score(Wolfe et al, 1999); (v) the 15-item Myasthenia Gravis Quality of Life (MG-QOL15) questionnaire (Burns et al, 2010); and (vi) the modified Rankin Scale (mRS) score. Assessments were done at least 10 h after the last dose of cholinesterase inhibitor.

### ELISA

Human anti-AChR antibody ELISA kit (RSR, ACE/96), Human anti-Titin antibody ELISA kit (DLD Diagnostika EA601/48), and Human anti-MuSK antibody ELISA (IBL, RE51021) were used for the detection for related autoantibodies, respectively. Human MIF DuoSet ELISA kit (R&D Systems, DY289) was used to measure its PBMC levels according to the manufacturer's instructions.

## Sample collection and processing

For single-cell analysis, freshly collected peripheral blood mononuclear cells (PBMCs) were acquired by density gradient centrifugation using Ficoll paque plus (Cytiva) in accordance with the manufacturer's instructions. Processed PBMC samples were subsequently used for single-cell RNA sequencing (scRNA-seq), scBCR-seq and scTCR-seq as detailed below. PBMCs were available from two MG patients at baseline, at 1 month and 3 months after infusion, as well as their matching infusion products.

CAR-T cells in the infusion products before transfer were processed for CITE-seq. Cells were blocked using antibody staining buffer which contained human TruStain FcX (Biolegend) and then stained with FITC-labeled human BCMA/TNFRSF17 protein (Acrobiosystems). After washing for three times, cells were then incubated with antibodies, anti-CD4, anti-CD8A, anti-FITC and isotype controls (Biolegend TotalSeq catalog 300567, 301071, 408311, 400283) for 30 min at 4 °C. Stained cells were washed three times and encapsulated for CITE-seq. Single-cell suspensions were loaded onto the 10x Chromium platform, followed by 5' gene expression library, immune repertoire library and cell surface protein library preparation.

## Library preparation and sequencing for scRNA-seq, scBCR-seq and scTCR-seq

Single-cell suspensions of infusion products and the blood samples collected from patients at baseline, at 1 month, and at 3 months after infusion were loaded onto the 10x Chromium platform for simultaneous transcriptome and BCR/TCR profiling. Transcriptome and BCR/TCR-enriched libraries were prepared for each sample following the manufacturer's instructions (10x Genomics). All libraries were then sequenced using a NovaSeq 6000 ((Illumina, San Diego, USA) with $2 \times 150$ paired-end reads to obtain sequence information.

## Data processing for scRNA-seq, and single-cell V(D)J sequencing

Raw scRNA-seq data were pre–processed (cellular barcode demultiplex, read alignment to the GRCh38 reference genome, data filtering, barcode and UMI counting, and identification of putative cells) with Cell Ranger v5.0.0 mkfastq and corresponding BCR and TCR reads were processed with Cell Ranger mkfastq and vdj.

## Seurat workflow of scRNA-seq and single-cell V(D)J sequencing

Downstream analyses were performed mainly using Seurat v4.2.0 (Hao et al, 2021). For quality control, cells with at least 200 genes, between 1000 and 40,000 UMI, novelty score (log10GenesPerUMI) above 0.7, and less than 10% mitochondrial RNA were retained for subsequent analysis. Genes detected in fewer than three cells were also filtered out. After data preprocessing, doublets classification and deleting were carried out for each sample using DoubletFinder v2.0.3 (McGinnis et al, 2019). Data derived from all enrolled samples were then merged into a single Seurat object, followed by log-normalization, HVGs identification, feature scaling, and principal–component analysis. Scores for G1, S, and G2/M cell cycle phases of cells were assigned on the basis of previously defined gene sets (Tirosh et al, 2016) using the CellCycleScoring function. To remove batch effects, we used harmony v0.1.0 (Korsunsky et al, 2019) with default parameters. The first 30 dimensions generated by Harmony were used to generate the neighborhood graph and embedded using UMAP. Clustering was performed using a resolution of 2.5 to determine the optimal number of clusters. Cell types for each of the clusters were defined by marker gene expression as shown in dot plot. For single-cell V(D)J sequencing, clonotype information was added to the metadata, and cells containing both BCR and TCR sequences were discarded for subsequent immune repertoire analyses. For TCR clonotypes tracking, overlaps among endogenous T cells at baseline, CAR-T cells in infusion products, and sorted CAR-T cells at 1 month post treatment were identified using their TRB amino acid sequences as previously described (Deng et al, 2020).

## Cellular interaction analysis

CellChat v1.4.0 was utilized to predict the cellular interaction between PB/PCs and other cell types following the official tutorial. In brief, a CellChat object was created using normalized counts of the expression matrix and overexpressed ligand-receptor interactions were identified. After projected onto the interaction network, data were then used to calculate the communication probabilities. Communications with less than ten cells were filtered out. Subsequently, probabilities of communications at the signaling pathway levels were calculated and aggregated networks were constructed with default parameters.

## Differentially expressed genes identification and pathway enrichment analysis

Differentially expressed genes of two comparison groups (B cells at baseline and B cells at 3-month post treatment) were identified

using FindAllMarkers function. Statistically significant differentially expressed genes were defined with *P*.adjust <0.05. Pathway enrichment analysis was performed using clusterProfile v4.4.2. Upregulated and downregulated differentially expressed genes were separately used as input, testing against the GO, KEGG and Reactome gene sets. Significant enrichment pathways were defined with *P*.adjust <0.05. In addition, inflammatory scoring for cells from patients at baseline, at 1 month and 3 months after the infusion was carried out with the AddModuleScore function in Seurat v4.2.0 using a literature gene set (Ren et al, 2021b). To illustrate differential activities among indicated cell groups, ssGSEA scores were calculated separately for each sample using package GSVA v1.44.0.

## Integrative analysis of scRNA-seq datasets from the literature

Gene expression matrices of previous published healthy donor-generated infusion products (Rodriguez-Marquez et al, 2022), BCMA–targeted infusion products (Li et al, 2021), CD19–targeted infusion products (axi-cel and tisa-cel) and corresponding PBMCs collected at baseline (Haradhvala et al, 2022) were acquired from the Gene Expression Omnibus (GEO accession GSE197851, GSE151310, GSE197268). Cells with less than 200 genes and more than 10% mitochondrial RNA were removed. Accordance with data processing pipeline described above, doublets filtering and harmony batch correction on the sample level were conducted to allow the identification of common populations. Differentially expressed genes of two nonpaired comparison groups were identified using FindAllMarkers function and were visualized with ggplot2. DEGs were then loaded into IPA software for core and comparison analysis to characterize the molecular and cellular function profiles of the two comparison groups. Activation or inhibition level of a specific canonical pathway was determined with the z score.

## Statistical analysis

Statistical tests used were indicated in the figure legends. No blinding or randomization was performed in our experiments. GraphPad Prism version 8.4.1 and R version 4.2.0 were used for statistical analysis.

## For more information

Website links for the antibodies used in the flow cytometry are following: PerCP/Cyanine5.5 anti-human CD45 Antibody (Biolegend, 304028): https://www.biolegend.com/en-us/products/percp-cyanine5-5-anti-human-cd45-antibody-4240; FITC anti-human CD3 antibody (BD Biosciences, 561802): https://www.bdbiosciences.com/zh-cn/products/reagents/flow-cytometry-reagents/research-reagents/single-color-antibodies-ruo/fitc-mouse-anti-human-cd3.561802; PE/Cyanine7 anti-Human CD4 (BD Biosciences, 560649): https://www.bdbiosciences.com/zh-cn/products/reagents/flow-cytometry-reagents/research-reagents/single-color-antibodies-ruo/pe-cy-7-mouse-anti-human-cd4.560649; APC/Cyanine7 anti-Human CD8 (BD Biosciences, 557834): https://www.bdbiosciences.com/zh-cn/products/reagents/flow-cytometry-reagents/research-reagents/single-color-antibodies-ruo/apc-cy-7-mouse-anti-human-cd8.557834; APC anti-human CD19 Antibody (Biolegend, 302212): https://www.biolegend.com/en-us/products/apc-anti-human-cd19-antibody-715; PE anti-human CD16 Antibody (Biolegend, 302056): https://www.biolegend.com/en-us/products/pe-anti-human-cd16-antibody-569; PE anti-human CD56 Antibody (Biolegend, 318306): https://www.biolegend.com/en-us/products/pe-anti-human-cd56-ncam-antibody-3796; FITC anti-Human CD38 (BD Biosciences, 567147): https://www.bdbiosciences.com/en-us/products/reagents/flow-cytometry-reagents/research-reagents/single-color-antibodies-ruo/fitc-mouse-anti-human-cd38.567147; PerCP/Cyanine5.5 anti-Human CD27 (BD Biosciences, 560612): https://www.bdbiosciences.com/zh-cn/products/reagents/flow-cytometry-reagents/research-reagents/single-color-antibodies-ruo/percp-cy-5-5-mouse-anti-human-cd27.560612; PerCP anti-Human CD45 (BD Biosciences, 347464): https://www.bdbiosciences.com/zh-cn/products/reagents/flow-cytometry-reagents/clinical-discovery-research/single-color-antibodies-ruo-gmp/percp-mouse-anti-human-cd45.347464; APC/Cyanine7 anti-human CD3 Antibody (Biolegend, 344818): https://www.biolegend.com/en-us/products/apc-cyanine7-anti-human-cd3-antibody-6940; FITC-labeled human BCMA Fc tag protein (Acrobiosystems, BCA-HF254): https://www.acrobiosystems.cn/P875-FITC-Labeled_Human_BCMA_%7C_TNFRSF17_Protein_Fc_Tag.html.

---

**The paper explained**

**Problem**
Approximately 10–20% of patients with MG are refractory to conventional first-line therapy. To date, several recent clinical cases have raised the possibility of CAR-T-cell therapy in the field of autoimmune diseases. However, these previous reports only offered short-term observation with median follow-up no more than 8 months.

**Results**
We reported here chimeric antigen receptor (CAR) T cells targeting B-cell maturation antigen (BCMA) in two patients with highly relapsed and refractory myasthenia gravis (MG) (one with AChR-IgG, and one with Musk-IgG). Both patients exhibited favorable safety profiles and persistent clinical improvements over 18 months. Notably, our data revealed a reconstitution of B-cell lineages with sustained reduced pathogenic autoantibodies might underlie the therapeutic efficacy. Moreover, single-cell transcriptomic and T-cell receptor combined sequencing analysis elucidated the temporal evolution of CAR-T cells and their behavior in vivo. By tracking the temporal evolution of CAR-T phenotypes, we demonstrated that proliferating cytotoxic-like CD8[+] T-cell clones were the main effectors in autoimmunity. Compared to CAR-T cells derived from healthy donors and lymphoma patients, CD8[+] T effector cells from MG patients displayed compromised cytotoxicity, diminished proliferation signature, and significant mitochondrial dysfunction before infusion. Moreover, CAR-T cells manufactured for MG patients exhibited defects post-manufacture, which might account for their unique characteristics. These findings could guide future studies to improve CAR-T-cell immunotherapy in autoimmune diseases.

**Impact**
The present case series illustrate great potentials of CAR-T-cell therapy in treating seropositive MG, with sustained depletion of autoantibodies and therapeutic efficacy beyond 18 months. Moreover, the single-cell analysis highlights distinct characteristics of CAR-T cells in patients with autoimmune diseases.

## Data availability

Raw sequencing data have been deposited in the Genome Sequence Archive in the National Genomics Data Center, China National Center for Bioinformation/Beijing Institute of Genomics, Chinese Academy of Sciences (GSA-Human: HRA004636) that could be publicly accessible at https://ngdc.cncb.ac.cn/search/?dbId=hra&q=HRA004636.

## Peer review information

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

## Acknowledgements

Our deepest gratitude goes first to Professor Jian-Feng Zhou, one of our sincere colleagues and also a major investigator in this project, who died from a heart attack early 2022. We appreciate him for his illuminating instruction and dedication to the project. Without him, this study could not have been initiated or reached its present form. We are also greatly indebted to all the patients who participated in the study, without whom this study would never have been accomplished. This study was supported by Ministry of Science and Technology China Brain Initiative Grant STI2030-Major Projects 2022ZD0204700 to Wei W; National Natural Science Foundation of China Grants 82071380, 81873743 to D-ST; National Natural Science Foundation of China Grant 82271341 to CQ; Knowledge Innovation Program of Wuhan Shuguang Project 2022020801020454 to CQ. The clinical trial was funded by Nanjing IASO Biotechnology Co., Ltd.

## Author contributions

**Dai-Shi Tian**: Conceptualization; Funding acquisition; Investigation; Writing—original draft; Project administration. **Chuan Qin**: Conceptualization; Funding acquisition; Investigation; Methodology; Writing—original draft; Project administration. **Ming-Hao Dong**: Investigation; Writing—original draft; Project administration. **Michael Heming**: Investigation; Methodology; Project administration; Writing—review and editing. **Luo-Qi Zhou**: Investigation; Project administration. **Wen Wang**: Supervision; Funding acquisition; Writing—review and editing. **Song-Bai Cai**: Supervision; Funding acquisition; Writing—review and editing. **Yun-Fan You**: Investigation; Project administration. **Ke Shang**: Investigation; Writing—original draft; Project administration. **Jun Xiao**: Investigation; Writing—original draft; Project administration. **Di Wang**: Conceptualization; Visualization; Methodology. **Chun-Rui Li**: Conceptualization; Visualization; Methodology; Writing—review and editing. **Min Zhang**: Conceptualization; Supervision; Visualization; Methodology; Writing—review and editing. **Bi-Tao Bu**: Supervision; Methodology; Writing—review and editing. **Gerd Meyer zu Hörste**: Conceptualization; Supervision; Writing—review and editing. **Wei Wang**: Conceptualization; Supervision; Writing—review and editing.

## Disclosure and competing interests statement

Wen Wang and Song-Bai Cai are employees of Nanjing IASO Biotechnology Co., Ltd and held interests in the company. Wen Wang is among the inventors

of patent applications related to the CT103A. The remaining authors declare no competing interests.

# Expanded View Figures

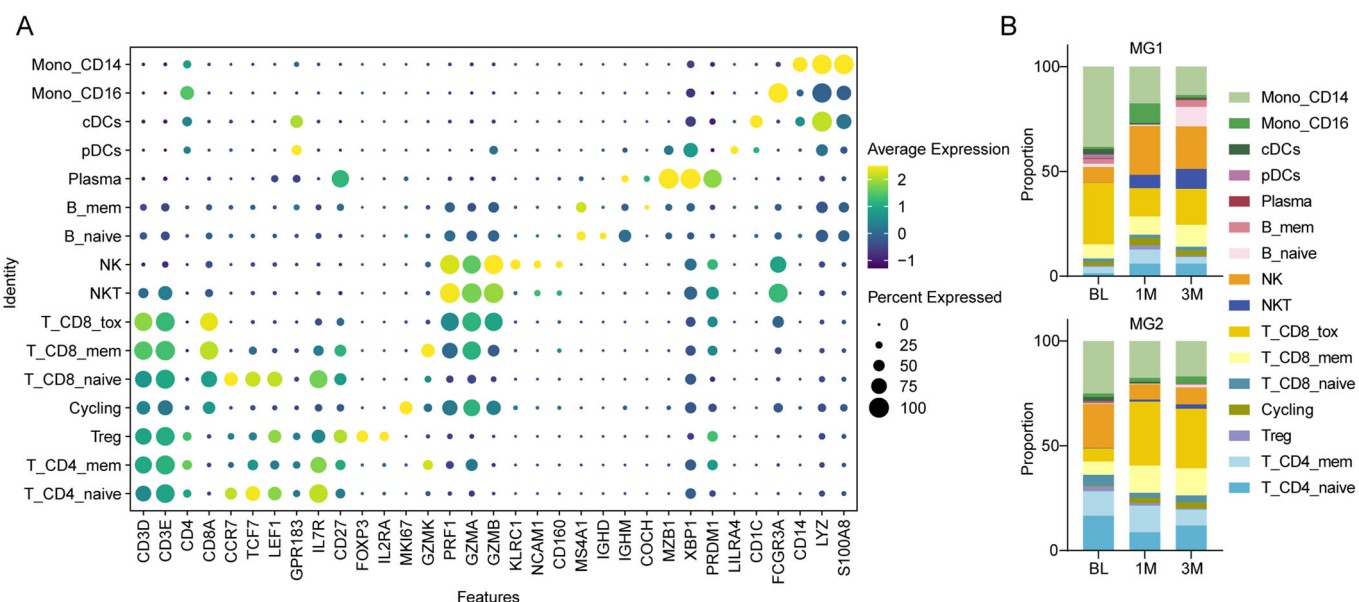

**Figure EV1. Additional single-cell transcriptional features of patients with myasthenia gravis treated with CAR-BCMA T-cell therapy.**

(A) Dot plot showing cell clusters denoted by gene expression of known markers. (B) Bar plots showing the frequency of cell subsets in individual patient at indicated time points.

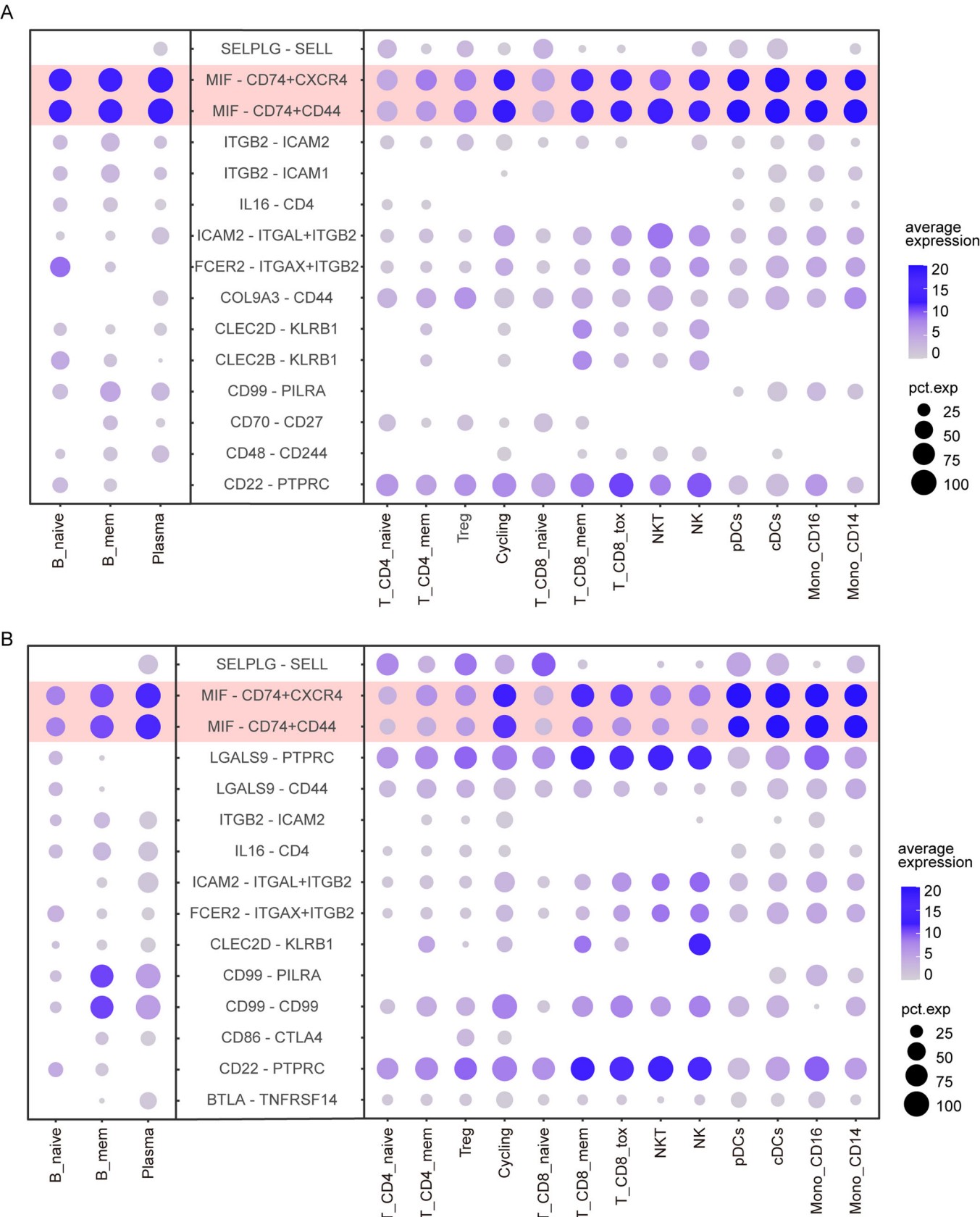

◀  **Figure EV2.  Additional ligand-receptor interaction analysis between immune cells.**

Gene expression dot plot of ligand-receptor expression. Left, expression in B-cell subsets; right, expression in other immune cell types. Shown are the Top15 interactions between B-cell subsets and other immune cell types with communication probabilities in (**A**) MG-1, (**B**) MG-2 at baseline.

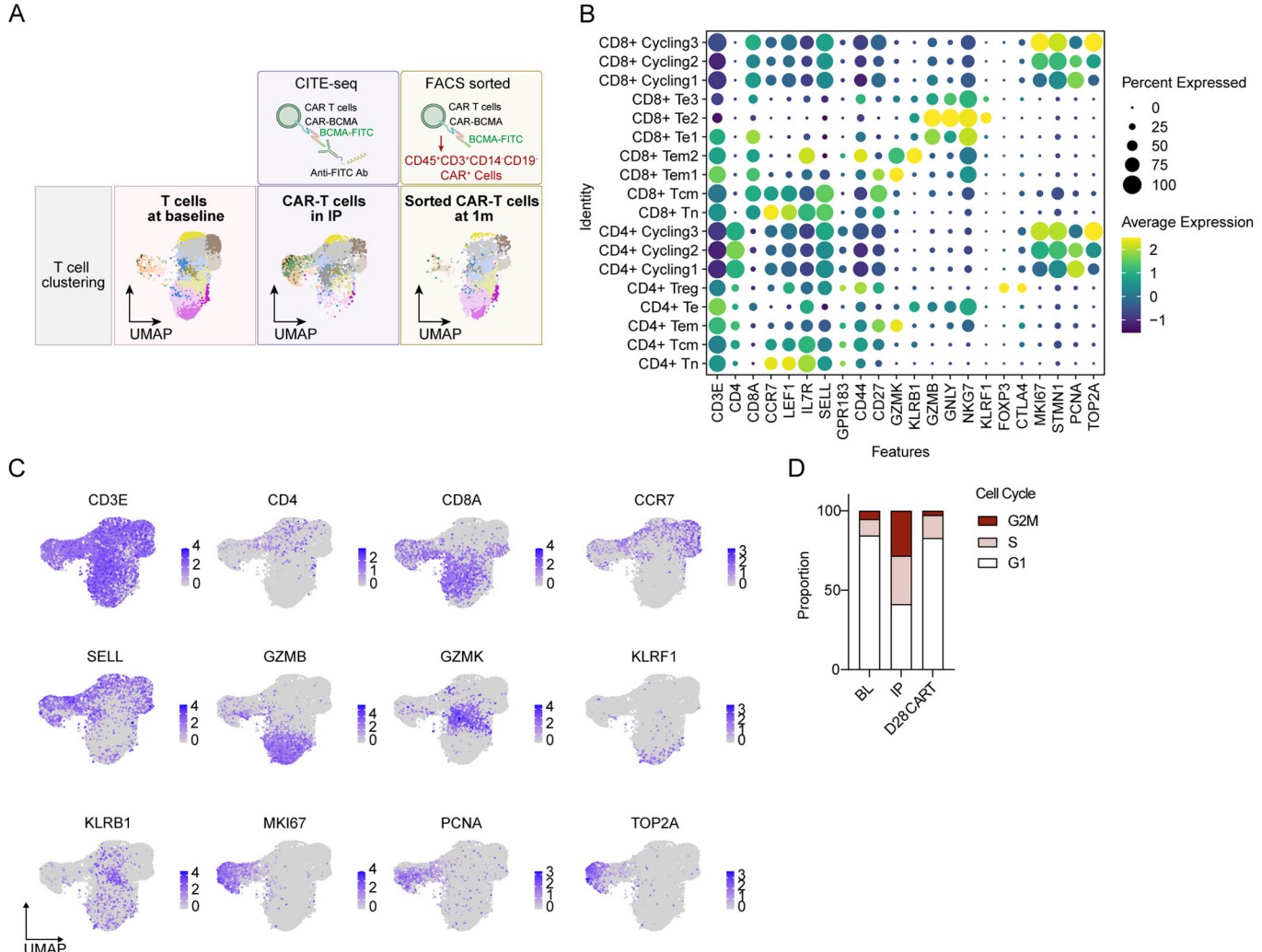

**Figure EV3. Expression of canonical markers identifies behavior of CAR-T cells in patients with myasthenia gravis.**

(A) Schematic illustration of cellular indexing of transcriptomes and epitopes by sequencing (CITE-seq) strategy used to detect the CAR on the T-cell surface, and flow cytometry to sort CAR-T cells in vivo at 1 month post infusion (see "Methods" for details). UMAP plots of T cells colored by different cell cluster and clone size. (B) Dot plots showing gene expression of known markers for T-cell and CAR T-cell subclusters. (C) UMAP plots indicating RNA expression of *CD3E, CD4, CD8A, CCR7, SELL, GZMB, GZMK, KLRF1, KLRB1, TOP2A, MKI67,* and *PCNA*. (D) Bar plots showing the frequency of cells in S phase or G2/M phase by cell cycle scoring (see "Methods" for details).

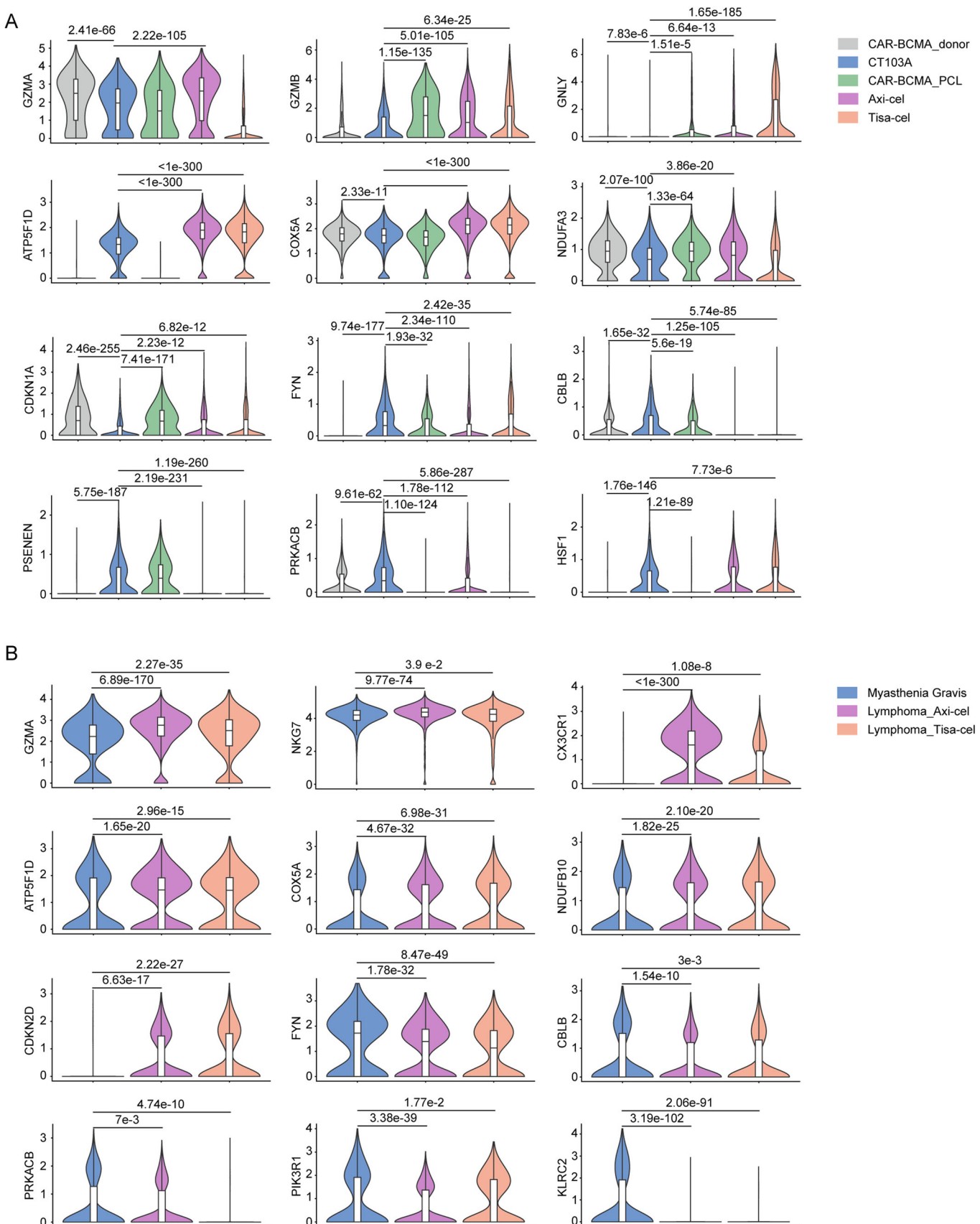

◀

**Figure EV4. Expression of indicated genes identifies characteristics of CD8 + Te cells (Baseline) and CD8+ cycling CAR-T cells (IP) in patients with myasthenia gravis.**

(A) Violin plots illustrating indicated genes in Fig. 5B, and representing the distribution of expression across each product. CT103A, $N = 4044$ cells; CAR-BCMA_donor, $N = 4739$ cells; CAR-BCMA_PCL, $N = 2111$ cells; Axi-cel, $N = 11,392$ cells; Tisa-cel, $N = 16,348$ cells. Boxes show median, Q1 and Q3 quartiles and whiskers up to 1.5× interquartile range. Pairwise comparisons were performed using a two-sided Wilcoxon rank-sum test with a Bonferroni correction. (B) Violin plots illustrating indicated genes in Fig. 5C, and representing the distribution of expression across each cohort. Myasthenia Gravis, $N = 2898$ cells; Lymphoma_Axi-cel, $N = 7123$ cells; Lymphoma_Tisa-cel, $N = 4857$ cells. Boxes show median, Q1 and Q3 quartiles and whiskers up to 1.5× interquartile range. Pairwise comparisons were performed using a two-sided Wilcoxon rank-sum test with a Bonferroni correction.

