## [Peer Review File · EMBO Molecular Medicine]

B cell lineage reconstitution underlies CAR-T cell therapeutic efficacy in patients with refractory myasthenia gravis

Dai-Shi Tian, Chuan Qin, Ming-Hao Dong, Michael Heming, Luo-Qi Zhou, Wen Wang, Song-Bai Cai, Yun-Fan You, Ke Shang, Jun Xiao, Di Wang, Chun-Rui Li, Min Zhang, Bi-Tao Bu, Gerd Meyer zu Hoerste, and Wei Wang

Corresponding authors: Wei Wang (wwang@tjh.tjmu.edu.cn) , Gerd Meyer zu Hoerste (gerd.meyezuhoerste@ukmuenster.de)

Review Timeline:

Submission Date:	23rd Sep 23
Editorial Decision:	23rd Nov 23
Revision Received:	20th Dec 23
Editorial Decision:	19th Jan 24
Revision Received:	6th Feb 24
Accepted:	7th Feb 24

Editor: Zeljko Durdevic

Transaction Report:

23rd Nov 2023

Dear Prof. Wang,

Thank you for the submission of your manuscript to EMBO Molecular Medicine, and please accept my apologies for the delay in getting back to you, which is due to the fact that one referee needed more time to complete his/her review. We have now received feedback from the three reviewers who agreed to evaluate your manuscript. While the referees #1 and #3 are overall supportive raising important but minor critique, referee #2 recognizes interest of the study but also raises concerns regarding the side effects of the CAR T-cell therapy particularly considering other available options with lower side effects. Based on the referee reports I would like to invite major revision of the present manuscript, with the understanding that all referee concerns should be addressed. If you would like to discuss further the points raised by the referees, I am available to do so via email or video. Let me know if you are interested in this option.

We would welcome the submission of a revised version within three months for further consideration. Please let us know if you require longer to complete the revision.

I look forward to receiving your revised manuscript.

Yours sincerely,

Zeljko Durdevic

We require:

- 1) A .docx formatted version of the manuscript text (including legends for main figures, EV figures and tables). Please make sure that the changes are highlighted to be clearly visible.
- 2) Individual production quality figure files as .eps, .tif, .jpg (one file per figure). For guidance, download the 'Figure Guide PDF': (<https://www.embopress.org/page/journal/17574684/authorguide#figureformat>).
- 3) A .docx formatted letter INCLUDING the reviewers' reports and your detailed point-by-point responses to their comments. As part of the EMBO Press transparent editorial process, the point-by-point response is part of the Review Process File (RPF), which will be published alongside your paper.
- 4) A complete author checklist, which you can download from our author guidelines (<https://www.embopress.org/page/journal/17574684/authorguide#submissionofrevisions>). Please insert information in the checklist that is also reflected in the manuscript. The completed author checklist will also be part of the RPF.
- 5) Please note that all corresponding authors are required to supply an ORCID ID for their name upon submission of a revised

manuscript.

6) It is mandatory to include a 'Data Availability' section after the Materials and Methods. Before submitting your revision, primary datasets produced in this study need to be deposited in an appropriate public database, and the accession numbers and database listed under 'Data Availability'. Please remember to provide a reviewer password if the datasets are not yet public (see <https://www.embopress.org/page/journal/17574684/authorguide#dataavailability>).

13) Author contributions: You will be asked to provide CRediT (Contributor Role Taxonomy) terms in the submission system. These replace a narrative author contribution section in the manuscript.

14) A Conflict of Interest statement should be provided in the main text.

Please also suggest a striking image or visual abstract to illustrate your article as a PNG file 550 px wide x 300-800 px high.

***** Reviewer's comments *****

Referee #1 (Remarks for Author):

BCMA-directed CAR-R treatment has already been applied to treat multiple myeloma, and another RNA-CAR-T therapy has been tested for refractory MG that did not use DNA vector or require lymphodepletion in patients.

Merit:

1. The current investigation represents another approach to test the clinical efficacy of the CAR-T treatment using DNA vectors and refractory MG patients.
2. The manuscript is well written, and the data presented in the manuscript are clear and sufficiently described.
3. The characterization of engineered cells after transfer and the RNA-seq analysis of targeted cells are all logical approaches.

There are some limitations of the study:

1. The study involved only one participant for each MG-type (AChR-IgG and MuSK-IgG), which affects the reliability of the results due to lack of statistical power. The results from one patient cannot be generalized for all MG patients. It undermines the purpose of the study, and any firm conclusions cannot be derived.
2. The study participants, who were refractory to prednisone, tacrolimus, and rituximab, discontinued the treatments only 1-3 days before starting CAR-T therapy. Thus, it is difficult to determine whether the clinical benefits observed in the two patients solely resulted from the CAR-T treatments or partly as an outcome from the previous treatments.
3. The authors found that the CAR-T-anti-BCMA treatment in patients reconstituted B cell lineages with naïve phenotypes. They need to clarify that statement as BCMA is present only on plasmablasts and plasma cells, but not on precursor B cells that differentiate to plasma cells.

Referee #2 (Comments on Novelty/Model System for Author):

The quality of the methodology is high as well as the importance of the research question.

Since this is not the first study of these characteristics in MG I see the impact as medium.

Additionally, I see the translation to the clinic low in its present form since there are other available treatment alternative which are less invasive.

Referee #2 (Remarks for Author):

In this manuscript, the authors claim that CAR-BCMA T cells are well-tolerated and highly effective in treating refractory MG, resulting in an improvement in physical function and serologic remission. Patient MG-1, with anti-AChR-IgG and anti-Titin antibodies, and patient MG-2, with anti-MuSK-IgG4, showed a poor response to first-line immunosuppressive therapies. Although B-cell-targeting therapy rituximab was considered an early therapeutic option, it also failed in these two patients. The present case series illustrates the potential of CAR-T cell therapy in treating seropositive MG, as evidenced by sustained depletion of autoantibodies and prolonged therapeutic efficacy beyond one year.

The reviewer has concerns regarding the choice of applying CAR T-cell therapy to these patients, as there are other clear options available with lower potential side effects, specifically targeting plasma cells and the effector functions of anti-AChR antibodies. Refer to Mané-Damas, Marina et al. "Novel treatment strategies for acetylcholine receptor antibody-positive

myasthenia gravis and related disorders" (Autoimmunity Reviews, vol. 21,7, 2022, 103104. doi:10.1016/j.autrev.2022.103104).

As is well known, CAR T-cell therapy is a type of immunotherapy that involves genetically modifying a patient's T cells to express a receptor targeting cancer cells. Neurological side effects, particularly neurotoxicity or cytokine release syndrome (CRS), can occur as a result of this therapy. The onset and duration of neurological side effects can vary widely among individuals. Neurological side effects from CAR T-cell therapy typically manifest within the first few days to weeks after treatment initiation. In some cases, they can occur within hours after the infusion of modified T cells. The duration of these side effects also varies. Some individuals may experience transient and mild symptoms, while others may face more severe and prolonged effects. The severity and timing of neurological side effects are often unpredictable. Symptoms may include confusion, delirium, seizures, and other neurological manifestations. Patients receiving CAR T-cell therapy should have regular follow-up appointments with their healthcare team to monitor for any delayed or long-term side effects.

It is crucial to consider if the potential side effects of this treatment are worthwhile, especially in neurological diseases like MG, when other treatment options are available in the market.

I would not consider such a complex approach, as depicted here, for the treatment of refractory MG patients a good current option. Instead, I would seek, if possible, more specific patient-tailored strategies, which will minimize the considerable risk of side effects, especially in the long term.

Referee #3 (Remarks for Author):

In this manuscript, Tian and colleagues reported very favorable outcome using chimeric antigen receptor (CAR) T cells targeting BCMA in two patients with refractory myasthenia gravis (one with AChR Ab, and one with MuSk Ab). They also utilized single cell RNA and TCR sequencing to track the temporal evolution of CAR-T phenotypes, and found a shift from autologous T effector (Te) to proliferating cytotoxic-like CD8 clones and NK-like Te cells post treatment in vivo. The clinical outcome and decline in antibody titers at 12 mo are significant and expected, but whether long-term remission will be induced remains to be determined. The work is exciting and very thorough and provides insight into immunologic events occurring after CAR-T cell therapy targeting BCMA in autoimmunity. Data presentation is excellent though may be too crowded /complex in the last 2 figures. Data interpretations are logical. Here are some questions /comments:

1. Cyclophosphamide has occasionally been used in refractory MG. Can the authors comment on the possibility that lymphodepletion prior to CAR-T cell infusion may have partially contributed to favorable outcome?
2. Since the number of targeted cells (BCMA+ B cells) in autoimmune diseases is much lower than in lymphomas or other cancers, is lymphodepletion allowing favorable condition for CAR-T cell proliferation necessary?
3. Can authors explain why do CAR-T cells persist only 30d in autoimmune disease compared to longer persistence in cancers?
4. In Fig. 1D on inflammatory mediators, do data represent transcript or protein levels?
5. In Fig. 3D legend, "The line width is proportional to the communication probability in NMOSD comparing to control group". I assume that the authors meant MG and not NMOSD.
6. Changes in MIF and serum BCMA levels have the potential to be used as biomarkers of response to therapy. Do authors have data on MIF and serum BCMA at 12 months post CAR-T cell therapy?
7. Fig. 4C legend can be expanded to indicate lower proliferation and energy metabolism and higher cytotoxicity at 1 mo compared with CAR-T cells in IP.
8. Mitochondrial dysfunction/oxidative phosphorylation abnormalities have been reported in SLE B lymphocytes (Takeshima et al 2022). Thus, impaired mitochondrial function in Te in this study may not be specific to myasthenia, and effect of prolonged exposure to immunosuppressants may be a contributing factor.

***** Reviewer's comments *****

Referee #1 (Remarks for Author):

BCMA-directed CAR-T treatment has already been applied to treat multiple myeloma, and another RNA-CAR-T therapy has been tested for refractory MG that did not use DNA vector or require lymphodepletion in patients.

Merit:

1. The current investigation represents another approach to test the clinical efficacy of the CAR-T treatment using DNA vectors and refractory MG patients.
2. The manuscript is well written, and the data presented in the manuscript are clear and sufficiently described.
3. The characterization of engineered cells after transfer and the RNA-seq analysis of targeted cells are all logical approaches.

Response: We appreciate the reviewer's affirmation and suggestions for our study, which guided us to revise this manuscript and strengthen our conclusion. Here we provided point-to-point response as shown below.

There are some limitations of the study:

1. The study involved only one participant for each MG-type (AChR-IgG and MU_sK-IgG), which affects the reliability of the results due to lack of statistical power. The results from one patient cannot be generalized for all MG patients. It undermines the purpose of the study, and any firm conclusions cannot be derived.

Response: We thank the reviewer for pointing out the limitations of our study. Indeed, we only reported here information on two patients before and after receiving CAR-T treatment. Although the number of examples is small, they have a strong representative significance. The patient MG-1 had typical AChR-IgG-positive myasthenia gravis. Her symptoms were not controlled under treatment with acetylcholinesterase inhibitor, prednisolone, and tacrolimus or rituximab. She had recurring crises before treatment and was unable to take care of herself. Another patient (MG-2) had typical MuSK-IgG-positive myasthenia gravis. Before enrolment, she had obvious symptoms in bulbar muscles, showing pharyngeal, and tongue weakness. After receiving CAR-T treatment, these two patients completely achieved drug-free clinical remission during a follow-up period of up to 18 months (our current follow-up period has been extended to 18 months, and these two patients completely stopped treatment with steroids, and all kinds of immunosuppressants). Currently only pyridostigmine is used, 30 mg twice a day for maintenance. These two cases mainly demonstrate the potentials of anti-BMCA CAR-T therapy in the two classic subtypes of myasthenia gravis.

We must admit that ONLY two patients suggesting that the prospects of CAR-T therapy have certain limitations, and we have now added corresponding descriptions in the limitations (**Page 13 Line 512-515 in marked version**). Thanks again for the constructive suggestions.

2. The study participants, who were refractory to prednisone, tacrolimus, and rituximab, discontinued the treatments only 1-3 days before starting CAR-T therapy. Thus, it is difficult to determine whether the clinical benefits observed in the two patients solely resulted from the

CAR-T treatments or partly as an outcome from the previous treatments.

Response: First, we thank the reviewer for pointing out this issue. We are very sorry that the lack of detailed description has caused some misunderstandings. Although both patients failed traditional treatment, the risk of recurrence or worsening that may arise from longer drug withdrawal is unethical. So, we only stopped oral steroids and immunosuppressants 1-3 days before lymphodepletion treatment. In the revised manuscript, we describe in detail the time points at which patients receive different treatments before CAR-T infusion and add related information to Figure 1. At the same time, both patients have been followed up for more than 18 months. We have also expanded the clinical follow-up data to 18 months and added it to the results and Figures 1-2 (**Page 7 Line 247-250 in marked version**). Within one and a half years after stopping steroids and immunosuppressants, both patients showed sustained clinical efficacy in relieving clinical symptoms. The sustained remission should have slight effects from previous treatments. We have written in the limitations (**Page 13 Line 515-518 in marked version**) the possible short-term effects of previous treatments. Thanks again for the reminders and help.

Revised Figure 1

Revised Figure 2

3. The authors found that the CAR-T-anti-BCMA treatment in patients reconstituted B cell lineages with naïve phenotypes. They need to clarify that statement as BCMA is present only on plasmablasts and plasma cells, but not on precursor B cells that differentiate to plasma cells.

Response: Thank you very much to the reviewer for such professional suggestions. We speculate that the anti-BCMA CAR-T cells could remove long-lived plasma cells and plasmablasts that secrete autoantibodies, and then the B cells that grow up again after lymphodepletion treatment do not differentiate into pathogenic plasma cells and plasmablasts. However, the mechanism underlying the reconstitution of B cell lineage is currently unknown and needs further exploration. We have added related description of the distribution of BCMA to the discussion (**Page 12 Line 462-468 in marked version**) and the reconstitution of B cells after anti-BCMA CAR-T cell therapy. Thank you again for all the constructive suggestions.

Referee #2 (Comments on Novelty/Model System for Author):

The quality of the methodology is high as well as the importance of the research question. Since this is not the first study of these characteristics in MG I see the impact as medium. additionally, I see the translation to the clinic low in its present form since there are other available treatment alternative which are less invasive.

Response: We are very grateful to the reviewer for pointing out the importance of our research. Regarding the comparison between our report and other articles on CAR-T treatment of MG, as well as the comparison with other “less invasive” treatments, we carefully summarized the relevant research progress and added a lot of comments in the discussion. Many thanks to the reviewer for the constructive suggestions.

Referee #2 (Remarks for Author):

In this manuscript, the authors claim that CAR-BCMA T cells are well-tolerated and highly effective in treating refractory MG, resulting in an improvement in physical function and serologic remission. Patient MG-1, with anti-AChR-IgG and anti-Titin antibodies, and patient MG-2, with anti-MuSK-IgG4, showed a poor response to first-line immunosuppressive therapies. Although B-cell-targeting therapy rituximab was considered an early therapeutic option, it also failed in these two patients. The present case series illustrates the potential of CAR-T cell therapy in treating seropositive MG, as evidenced by sustained depletion of autoantibodies and prolonged therapeutic efficacy beyond one year.

Response: Thanks again to the reviewer for the encouragement of the highlights of our study. We will revise and improve it point-by-point based on your valuable comments.

The reviewer has concerns regarding the choice of applying CAR T-cell therapy to these patients, as there are other clear options available with lower potential side effects, specifically targeting plasma cells and the effector functions of anti-AChR antibodies. Refer to Mané-Damas, Marina et al. "Novel treatment strategies for acetylcholine receptor antibody-positive myasthenia gravis and related disorders" (Autoimmunity Reviews, vol. 21,7, 2022, 103104. doi:10.1016/j.autrev.2022.103104).

Response: Thanks to the reviewer's suggestions. We have thoroughly read this article on the progress of MG treatment and cited it as an important reference (Mané-Damas *et al*, 2022) (Ref.27). Indeed, there are currently increasing innovative strategies for the treatment of MG were under evaluation in various clinical trials. And novel strategies like complement inhibitors and FcRn antagonists were approved by FDA since 2017. However, minimal options are officially approved and commercially available treatments in mainland China before 2023. Our two patients were enrolled and received CAR-T treatment in the early 2022. At that time, there was only rituximab as off-label choice for refractory myasthenia gravis in China.

There are two other teams working on CAR-T in treating MG patients. One performed RNA-engineered anti-BCMA CAR-T treatment in MG patients without lymphodepletion therapy (Granit *et al*, 2023). This protocol requires repeated administration (twice weekly for 3 weeks, once weekly for 6 weeks, or once monthly for 6 months in Part 2) and has shown maintained improvements during a 5-months follow-up in all participants who received weekly infusions for 6

weeks. Another case report recently published on *Lancet Neuro* was an anti-CD19 CAR-T treatment (conventional DNA-engineered) given in one patient early this year (2023) (Haghikia *et al*, 2023). Clinical efficacy and primary safety were observed in the first 62 days after CAR-T treatment.

In our report, both the two patients showed clinical remission for up to 18 months after one dose of CAR-T infusion, which provides a relatively long period of observation and gives many hints on safety and long-term efficacy.

We hope that our case report could help neurologists around the world learn more about the possibilities of such treatments in autoimmune diseases. Meanwhile, we have now added a lot of related comments in the discussion (**Page 11 Line 406-431 in marked version**). Thanks again to the reviewers for their constructive suggestions.

As is well known, CAR T-cell therapy is a type of immunotherapy that involves genetically modifying a patient's T cells to express a receptor targeting cancer cells. Neurological side effects, particularly neurotoxicity or cytokine release syndrome (CRS), can occur as a result of this therapy. The onset and duration of neurological side effects can vary widely among individuals. Neurological side effects from CAR T-cell therapy typically manifest within the first few days to weeks after treatment initiation. In some cases, they can occur within hours after the infusion of modified T cells. The duration of these side effects also varies. Some individuals may experience transient and mild symptoms, while others may face more severe and prolonged effects. The severity and timing of neurological side effects are often unpredictable. Symptoms may include confusion, delirium, seizures, and other neurological manifestations. Patients receiving CAR T-cell therapy should have regular follow-up appointments with their healthcare team to monitor for any delayed or long-term side effects.

Response: We thank the reviewer for pointing out this issue.

According to the available clinical trials of CAR-T therapy on autoimmune diseases and hematological malignancies, most adverse effects occurred in the early stage. To date, no prolonged or delayed adverse events have been reported in the treatment of autoimmune diseases (Mackensen *et al*, 2022; Muller *et al*, 2023; Qin *et al*, 2023). While in hematological tumors, several cases of prolonged toxicity did appear. For instance, two patients with multiple myeloma had developed a grade 3 cerebellar disorder that was considered to be a dose-limiting-toxicity effect for the dose of 450×10^6 CAR-T-anti-GPRC5D at 6.5 and 8.4 months, respectively (Mailankody *et al*, 2022). Another clinical trial using anti-BCMA CAR-T cells, the patient developed movement disorder with features of parkinsonism at day 101 after CAR-T cell infusion (Van Oekelen *et al*, 2021). These two neurological adverse events were both speculated to be related with target antigens expressing in the brain tissue. Additionally, prolonged grade 3/4 cytopenia such as neutropenia, thrombocytopenia, anemia and lymphocytopenia, were also reported in several patients, with the susceptibility to develop fatal infectious and hemorrhagic events (Benjamin *et al*, 2020; Mei *et al*, 2021; Munshi *et al*, 2021).

In our study, we also pay a lot of attention on side effects of CAR-T therapy, including CRS, ICANS, neurotoxicity, and etc. Fortunately, following treatment with CAR-T cells, patient MG-1 experienced transient grade 1 CRS (pyrexia with maximum temperature 39.3 °C) at day 8, which resolved within 1 day automatically. Patient MG-2 experienced no CRS. Neutropenia and lymphocytopenia were observed within 1-month post-infusion, and all resolved within 4 weeks.

No immune effector cell-associated neurotoxicity syndrome, other neurologic toxic effects, or dose-limiting-toxicity were observed. All AEs were documented, and regular follow-up were performed every 3 months. Currently, both patients have been followed up for more than 18 months and no delayed adverse effects were observed. Nonetheless, subsequent regular follow-up and continuous monitoring are still needed for the prevention of any potential delayed or long-term side effects. We have also added the subsequent follow-up information to the **revised Figure 1-2**.

As suggested, a detailed discussion of the potential side effects that CAR-T therapy may bring and the subsequent monitoring of delayed or long-term side effects has now been added to the Discussion section (**Page 11-12 Line 432-457 in marked version**).

Thank you again for the professional advice.

Revised Figure 1

Revised Figure 2

It is crucial to consider if the potential side effects of this treatment are worthwhile, especially in neurological diseases like MG, when other treatment options are available in the market. I would not consider such a complex approach, as depicted here, for the treatment of refractory MG patients a good current option. Instead, I would seek, if possible, more specific patient-tailored strategies, which will minimize the considerable risk of side effects, especially in the long term.

Response: Thanks again to the reviewer for these professional concerns. As doctors specializing in neuroimmunology, we hope that more treatment options could be explored for patients with refractory myasthenia gravis and subsequent frustration. At the same time, we also want to point out that the sustained clinical efficacy for over 18 months following one single dose of CAR-T cell infusion is really exciting. Both the patients really enjoy their drug-free life (all regular administration of oral steroids and all kinds of immunosuppressants were stopped before CAR-T infusion).

For now, the exploration of CAR-T in treating autoimmune diseases is evolving in various research centers around the world (Schett *et al*, 2023). In addition to the promising clinical effects of CAR-T in treating MG, our study also explored the molecular mechanism of CAR-T cells in treating autoimmune diseases through single-cell transcriptome analysis. Different characteristics of CAR-T cells in autoimmune diseases and those cells in hematological tumors were depicted. It is hoped that through the description of these molecular characteristics, our findings could provide some evidence for more specific patient-tailored CAR-T strategies in treating patients with autoimmune diseases in the future.

In addition, CAR-T therapy with more specific patient-tailored strategies, such as MuSK-CAAR T cell technology (Oh *et al*, 2023), and NMDAR-CAAR T cell therapy (Reincke *et al*, 2023), were under evaluation in several preclinical studies, while more clinical evidence is warranted. We have now added related comments in the discussion (**Page 12 Line 456-457 in marked version**).

Thank you again for all the professional advices and constructive help.

Referee #3 (Remarks for Author):

In this manuscript, Tian and colleagues reported very favorable outcome using chimeric antigen receptor (CAR) T cells targeting BCMA in two patients with refractory myasthenia gravis (one with AChR Ab, and one with MuSk Ab). They also utilized single cell RNA and TCR sequencing to track the temporal evolution of CAR-T phenotypes, and found a shift from autologous T effector (Te) to proliferating cytotoxic-like CD8 clones and NK-like Te cells post treatment in vivo. The clinical outcome and decline in antibody titers at 12 mo are significant and expected, but whether long-term remission will be induced remains to be determined. The work is exciting and very thorough and provides insight into immunologic events occurring after CAR-T cell therapy targeting BCMA in autoimmunity. Data presentation is excellent though may be too crowded /complex in the last 2 figures. Data interpretations are logical. Here are some questions /comments:

Response: We appreciate the reviewer's affirmation and suggestions for our study, which guided us to revise this manuscript and strengthen our conclusion. Thank you for pointing out the issue about the data presentation. We've divided Fig 4 into two individual figures (Fig 4 and Fig 5 in the revised version) and appropriately reduced the numbers of genes labeled in the volcano plots of Fig 5 (Fig 6 in the revised version) (Page 20 Line 767-783 in marked version).

Moreover, we also treasured your following construction suggestions, and we've provided point-to-point response as shown below.

Revised Figure 4

Revised Figure 5

Revised Figure 6

1. Cyclophosphamide has occasionally been used in refractory MG. Can the authors comment on the possibility that lymphodepletion prior to CAR-T cell infusion may have partially contributed to favorable outcome?

Response: We thank the reviewer for pointing out this issue. Indeed, cyclophosphamide has occasionally been used in refractory MG, which may attribute partially to clinical efficacy we observed in these two patients. In addition, lymphodepletion therapy we adopted in this study include the use of fludarabine, which was not used in treating MG. We cannot exclude the effects of cyclophosphamide and fludarabine, while the effects were supposed not to persist over 3 months. At the same time, both patients have been followed up for more than 18 months. We have also expanded the clinical follow-up data to 18 months and added it to the results and Figures 1-2

(Page 7 Line 247-250 in marked version). In our 18-month follow-up observation, we found that the two patients still showed sustained clinical efficacy. This is inspiring. We have now added related comments as limitations in the discussion (Page 13 Line 515-518 in marked version). Thank you again for all the constructive suggestions.

Revised Figure 1

Revised Figure 2

2. Since the number of targeted cells (BCMA+ B cells) in autoimmune diseases is much lower than in lymphomas or other cancers, is lymphodepletion allowing favorable condition for CAR-T cell proliferation necessary?

Response: Thank you to the reviewer for raising such a question worthy of deep consideration. We had similar doubts before designing the trial protocol.

However, through literature search, we found that CAR-T cell therapy in some tumors has obvious limitations in its expansion without lymphodepletion (Geyer *et al*, 2019; Hirayama *et al*, 2019; Narayan *et al*, 2022; Turtle *et al*, 2016). We also noticed that, in a recent report (Granit *et al*, 2023), RNA-engineered CAR-T cells were administered to MG patients without lymphodepletion chemotherapy. But at the same time, patients were given repeated infusions of CAR-T cells (twice weekly for 3 weeks, once weekly for 6 weeks, or once monthly for 6 months in Part 2). In addition, preclinical studies on CAR-T treating autoimmune diseases also adopted lymphodepletion

treatment for allowing favorable condition for CAR-T cells (Jin *et al*, 2021; Kansal *et al*, 2019) while CAR-CTLs injected without preconditioning lymphodepletion were barely detectable by day 15 in any of the analyzed tissues (Kobayashi *et al*, 2020). Collectively, lymphodepletion might be necessary for CAR-T cell treatment, benefiting the homeostasis and expansion of transferred CAR-T cells. Therefore, we conducted lymphodepletion before CAR-T treatment to assure CAR-T cells' successfully expanding. We have added related comments in the revised discussion (**Page 11 Line 421-424 in marked version**). Thanks again for all the constructive suggestions.

3. Can authors explain why do CAR-T cells persist only 30d in autoimmune disease compared to longer persistence in cancers?

Response: We totally agreed with the reviewer. The shorter persistence of CAR-T cells (approximately 30 days) in autoimmune diseases compared to that in hematological cancers, was observed in early reports (Mackensen *et al.*, 2022; Qin *et al.*, 2023) and the present study. Unlike cancers, circulating B cells/plasma cells are easily and rapidly cleared, resulting in limited stimulation of functional effector T cells that persist after targeting cell eradication. In addition, our study using single-cell transcriptomic analysis highlights the importance of infusing proliferating CAR⁺ T_H cells manufactured from endogenous T_H phenotype in the final expansion stage of autoimmunity. The suppressed effector signature and profound mitochondrial dysfunction of T_H cells in patients with refractory MG, which might result from multiple/long-term immunosuppressants and steroid use prior to enrolment, and subsequent compromised properties of the manufactured CAR-T cells, provide a possible explanation for their poor persistence in autoimmune diseases, in addition to the relatively low levels of antigen stimulation compared with malignancies. Additionally, inhibited proliferating properties and enhanced cell exhaustion/dysfunction were also observed in basal cells and manufactured CAR-T cells of MG patients, which might also lead to their relatively shorter persistence and poorer effectiveness, in addition to the limited antigen exposure, enriched cell types, and suppressed effector and profound mitochondrial dysfunction. We have added related comments in the revised discussion (**Page 13 Line 502-511 in marked version**).

4. In Fig. 1D on inflammatory mediators, do data represent transcript or protein levels?

Response: We thank the reviewer for pointing out this issue. Data in Fig 1D represent protein levels of inflammatory mediators. We've added related description in the revised result and figure legend (**Page 6 Line 203 in marked version**).

5. In Fig. 3D legend, "The line width is proportional to the communication probability in NMOSD comparing to control group". I assume that the authors meant MG and not NMOSD.

Response: We thank the reviewer for pointing out this issue. We've corrected this mistake in Fig 3D legend (**Page 20 Line 752 in marked version**).

6. Changes in MIF and serum BCMA levels have the potential to be used as biomarkers of response to therapy. Do authors have data on MIF and serum BCMA at 12 months post CAR-T cell therapy?

Response: We thank the reviewer for pointing out this issue. We did test the serum levels of MIF

and sBCMA at 12 months and 18 months post infusion. Consistent with the percentage of plasmablasts and plasma cells and the Ig levels in blood, the serum MIF and sBCMA persisted at a relatively low levels post infusion.

7. Fig. 4C legend can be expanded to indicate lower proliferation and energy metabolism and higher cytotoxicity at 1 mo compared with CAR-T cells in IP.

Response: We thank the reviewer for pointing out this issue. We've expanded the Fig 4C legend (Page 20 Line 771-773 in marked version) to make it clearly understood.

8. Mitochondrial dysfunction/oxidative phosphorylation abnormalities have been reported in SLE B lymphocytes (Takeshima et al 2022). Thus, impaired mitochondrial function in Te in this study may not be specific to myasthenia, and effect of prolonged exposure to immunosuppressants may be a contributing factor.

Response: Thank you very much to the reviewer for such constructive suggestions. We totally agreed with the reviewer. Takeshima et al reported the mitochondrial abnormalities of B lymphocytes in SLE, similar with the profound mitochondrial dysfunction and suppressed effector signature of Te cells in patients with refractory MG in our study, which might partially result from multiple/long-term immunosuppressants and steroid use prior to enrolment. We have cited this study as an important reference (Ref.41), and added relevant comments in Discussion as suggested (Page 13 Line 500-502 in marked version). We thank the reviewer again for this great suggestion to strengthen our conclusion.

References

Benjamin R, Graham C, Yallop D, Jozwik A, Mirci-Danicar OC, Lucchini G, Pinner D, Jain N, Kantarjian H, Boissel N *et al* (2020) Genome-edited, donor-derived allogeneic anti-CD19 chimeric antigen receptor T cells in paediatric and adult B-cell acute lymphoblastic leukaemia: results of two phase 1 studies. *Lancet* 396: 1885-1894

Geyer MB, Riviere I, Senechal B, Wang X, Wang Y, Purdon TJ, Hsu M, Devlin SM, Palomba ML, Halton E *et al* (2019) Safety and tolerability of conditioning chemotherapy followed by CD19-targeted CAR T cells for relapsed/refractory CLL. *JCI Insight* 5

Granit V, Benatar M, Kurtoglu M, Miljkovic MD, Chahin N, Sahagian G, Feinberg MH, Slansky A, Vu T, Jewell CM *et al* (2023) Safety and clinical activity of autologous RNA chimeric antigen receptor T-cell therapy in myasthenia gravis (MG-001): a prospective, multicentre, open-label, non-randomised phase 1b/2a study. *Lancet Neurol* 22: 578-590

Haghikia A, Hegelmaier T, Wolleschak D, Bottcher M, Desel C, Borie D, Motte J, Schett G, Schroers R, Gold R *et al* (2023) Anti-CD19 CAR T cells for refractory myasthenia gravis. *Lancet Neurol* 22: 1104-1105

Hirayama AV, Gauthier J, Hay KA, Voutsinas JM, Wu Q, Gooley T, Li D, Cherian S, Chen X, Pender BS *et al* (2019) The response to lymphodepletion impacts PFS in patients with aggressive non-Hodgkin lymphoma treated with CD19 CAR T cells. *Blood* 133: 1876-1887

Jin X, Xu Q, Pu C, Zhu K, Lu C, Jiang Y, Xiao L, Han Y, Lu L (2021) Therapeutic efficacy of anti-CD19 CAR-T cells in a mouse model of systemic lupus erythematosus. *Cell Mol Immunol* 18: 1896-1903

Kansal R, Richardson N, Neeli I, Khawaja S, Chamberlain D, Ghani M, Ghani QU, Balazs L, Beranova-Giorgianni S, Giorgianni F *et al* (2019) Sustained B cell depletion by CD19-targeted

CAR T cells is a highly effective treatment for murine lupus. *Sci Transl Med* 11

Kobayashi S, Thelin MA, Parrish HL, Deshpande NR, Lee MS, Karimzadeh A, Niewczas MA, Serwold T, Kuhns MS (2020) A biomimetic five-module chimeric antigen receptor ((5M)CAR) designed to target and eliminate antigen-specific T cells. *Proc Natl Acad Sci U S A* 117: 28950-28959

Mackensen A, Muller F, Mougiakakos D, Boltz S, Wilhelm A, Aigner M, Volkl S, Simon D, Kleyer A, Munoz L *et al* (2022) Anti-CD19 CAR T cell therapy for refractory systemic lupus erythematosus. *Nat Med* 28: 2124-2132

Mailankody S, Devlin SM, Landa J, Nath K, Diamonte C, Carstens EJ, Russo D, Auclair R, Fitzgerald L, Cadzin B *et al* (2022) GPRC5D-Targeted CAR T Cells for Myeloma. *N Engl J Med* 387: 1196-1206

Mané-Damas M, Molenaar PC, Ulrichs P, Marcuse F, De Baets MH, Martinez-Martinez P, Losen M (2022) Novel treatment strategies for acetylcholine receptor antibody-positive myasthenia gravis and related disorders. *Autoimmunity Reviews* 21

Mei H, Li C, Jiang H, Zhao X, Huang Z, Jin D, Guo T, Kou H, Liu L, Tang L *et al* (2021) A bispecific CAR-T cell therapy targeting BCMA and CD38 in relapsed or refractory multiple myeloma. *J Hematol Oncol* 14: 161

Muller F, Boeltz S, Knitza J, Aigner M, Volkl S, Kharboutli S, Reimann H, Taubmann J, Kretschmann S, Rosler W *et al* (2023) CD19-targeted CAR T cells in refractory antisynthetase syndrome. *Lancet* 401: 815-818

Munshi NC, Anderson LD, Jr., Shah N, Madduri D, Berdeja J, Lonial S, Raje N, Lin Y, Siegel D, Oriol A *et al* (2021) Idecabtagene Vicleucel in Relapsed and Refractory Multiple Myeloma. *N*

Engl J Med 384: 705-716

Narayan V, Barber-Rotenberg JS, Jung IY, Lacey SF, Rech AJ, Davis MM, Hwang WT, Lal P, Carpenter EL, Maude SL *et al* (2022) PSMA-targeting TGFbeta-insensitive armored CAR T cells in metastatic castration-resistant prostate cancer: a phase 1 trial. *Nat Med* 28: 724-734

Oh S, Mao X, Manfredo-Vieira S, Lee J, Patel D, Choi EJ, Alvarado A, Cottman-Thomas E, Maseda D, Tsao PY *et al* (2023) Precision targeting of autoantigen-specific B cells in muscle-specific tyrosine kinase myasthenia gravis with chimeric autoantibody receptor T cells.

Nat Biotechnol 41: 1229-1238

Qin C, Tian DS, Zhou LQ, Shang K, Huang L, Dong MH, You YF, Xiao J, Xiong Y, Wang W *et al* (2023) Anti-BCMA CAR T-cell therapy CT103A in relapsed or refractory AQP4-IgG seropositive neuromyelitis optica spectrum disorders: phase 1 trial interim results. *Signal*

Transduct Target Ther 8: 5

Reincke SM, von Wardenburg N, Homeyer MA, Kornau HC, Spagni G, Li LY, Kreye J, Sanchez-Sendin E, Blumenau S, Stappert D *et al* (2023) Chimeric autoantibody receptor T cells deplete NMDA receptor-specific B cells. *Cell* 186: 5084-5097 e5018

Schett G, Mackensen A, Mougiakakos D (2023) CAR T-cell therapy in autoimmune diseases.

Lancet 402: 2034-2044

Turtle CJ, Hanafi LA, Berger C, Hudecek M, Pender B, Robinson E, Hawkins R, Chaney C, Cherian S, Chen X *et al* (2016) Immunotherapy of non-Hodgkin's lymphoma with a defined ratio of CD8+ and CD4+ CD19-specific chimeric antigen receptor-modified T cells. *Sci Transl*

Med 8: 355ra116

Van Oekelen O, Aleman A, Upadhyaya B, Schnakenberg S, Madduri D, Gavane S,

Teruya-Feldstein J, Crary JF, Fowkes ME, Stacy CB *et al* (2021) Neurocognitive and hypokinetic movement disorder with features of parkinsonism after BCMA-targeting CAR-T cell therapy. *Nat Med* 27: 2099-2103

19th Jan 2024

Dear Prof. Wang,

Thank you for the submission of your revised manuscript to EMBO Molecular Medicine. We have now heard back from the one referee who agreed to evaluate your manuscript. This referee also assessed author responses to concerns raised by other referees. I am pleased to inform you that we will be able to accept your manuscript pending the following final amendments:

1) We note that you currently have, a total of 3 first authors. Is that correct? Do you confirm equal contribution of these authors, able to take full responsibility for the paper and its content? While there is no limit per se to the number of first authors, 3 authors is rather rare, and may not reflect as intended to the community.

2) Formatting: Please correct order of the manuscript sections: Abstract / Introduction / Results / Discussion / Materials and Methods / Data Availability / Acknowledgements / Disclosure and competing interests statement / The Paper Explained / For More Information / References / Figure legends / Tables and their legends / Expanded View Figure legends

3) In the main manuscript file, please do the following:

- Please address all comments suggested by our data editors listed below:

o Figure legends:

1. Please indicate the statistical test used for data analysis in the legends of figures 3f; 6b-c.

2. Please note that the box plot needs to be defined in terms of minima, maxima, centre, bounds of box and whiskers, and percentile in the legend of figure 3e.

3. Please note that information related to n is missing in the legends of figures 3e; 6b-c; EV 4a-b.

- Updated the callouts of EV figures in the text to Fig EV1-4.

- In M&M, provide the statement that the experiments conformed to the WMA Declaration of Helsinki and to the principles set out in the Department of Health and Human Services Belmont Report.

- In M&M, add a statistical paragraph that should reflect all information that you have filled in the Authors Checklist, especially regarding randomization, blinding, replication.

- Please rename "Potential Conflicts of Interest" to "Disclosure Statement & Competing Interests". We updated our journal's competing interests policy in January 2022 and request authors to consider both actual and perceived competing interests. Please review the policy <https://www.embopress.org/competing-interests> and update your competing interests if necessary.

- Author contributions: Please remove it from the manuscript and specify author contributions in our submission system. CRediT has replaced the traditional author contributions section because it offers a systematic machine-readable author contributions format that allows for more effective research assessment. You are encouraged to use the free text boxes beneath each contributing author's name to add specific details on the author's contribution. More information is available in our guide to authors:

<https://www.embopress.org/page/journal/17574684/authorguide#authorshipguidelines>

- Data availability: Please make sure that all data deposited in public repositories are freely accessible upon publication. Also, please check provided URLs and make sure that the link is correct.

4) Appendix: Please move all supplementary material and methods to the main manuscript file.

5) Funding: Please merge it with the "Acknowledgements" and make sure that information about all sources of funding are complete in both our submission system and in the manuscript. The Nanjing IASO Therapeutics Ltd is currently missing in our submission system.

6) The Paper Explained: Please rename "Research in context" to "The Paper Explained" and add it to the main manuscript file.

7) Synopsis: Every published paper now includes a "Synopsis" to further enhance discoverability. Synopses are displayed on the journal webpage and are freely accessible to all readers. They include separate synopsis image and synopsis text.

- Synopsis image: Please provide a striking image or visual abstract as a high-resolution jpeg file 550 px-wide x (250-400)-px high to illustrate your article.

- Synopsis text: Please provide a short standfirst (maximum of 300 characters, including space) as well as 2-5 one sentence bullet points that summarise the paper as a .doc file. Please write the bullet points to summarise the key NEW findings. They should be designed to be complementary to the abstract - i.e. not repeat the same text. We encourage inclusion of key acronyms and quantitative information (maximum of 30 words / bullet point). Please use the passive voice.

8) For more information: This space should be used to list relevant web links for further consultation by our readers. Could you identify some relevant ones and provide such information as well? Some examples are patient associations, relevant databases, OMIM/proteins/genes links, author's websites, etc...

9) As part of the EMBO Publications transparent editorial process initiative (see our Editorial at

<http://embomolmed.embopress.org/content/2/9/329>), EMBO Molecular Medicine will publish online a Review Process File (RPF) to accompany accepted manuscripts. This file will be published in conjunction with your paper and will include the anonymous referee reports, your point-by-point response and all pertinent correspondence relating to the manuscript. Let us know whether you agree with the publication of the RPF and as here, if you want to remove or not any figures from it prior to publication. Please note that the Authors checklist will be published at the end of the RPF.

10) Please provide a point-by-point letter INCLUDING my comments as well as the reviewer's reports and your detailed

responses (as Word file).

I look forward to reading a new revised version of your manuscript as soon as possible.

Yours sincerely,

Zeljko Durdevic

*** Instructions to submit your revised manuscript ***

- 1) a .docx formatted version of the manuscript text (including Figure legends and tables)
- 2) Separate figure files*
- 3) supplemental information as Expanded View and/or Appendix. Please carefully check the authors guidelines for formatting Expanded view and Appendix figures and tables at <https://www.embopress.org/page/journal/17574684/authorguide#expandedview>
- 4) a letter INCLUDING the reviewer's reports and your detailed responses to their comments (as Word file).
- 5) The paper explained: EMBO Molecular Medicine articles are accompanied by a summary of the articles to emphasize the major findings in the paper and their medical implications for the non-specialist reader. Please provide a draft summary of your article highlighting
 - the medical issue you are addressing,
 - the results obtained and
 - their clinical impact.This may be edited to ensure that readers understand the significance and context of the research. Please refer to any of our published articles for an example.
- 6) For more information: There is space at the end of each article to list relevant web links for further consultation by our readers. Could you identify some relevant ones and provide such information as well? Some examples are patient associations, relevant databases, OMIM/proteins/genes links, author's websites, etc...
- 7) Author contributions: the contribution of every author must be detailed in a separate section.
- 8) EMBO Molecular Medicine now requires a complete author checklist (<https://www.embopress.org/page/journal/17574684/authorguide>) to be submitted with all revised manuscripts. Please use the checklist as guideline for the sort of information we need WITHIN the manuscript. The checklist should only be filled with page numbers where the information can be found. This is particularly important for animal reporting, antibody dilutions (missing) and

exact values and n that should be indicated instead of a range.

9) Every published paper now includes a 'Synopsis' to further enhance discoverability. Synopses are displayed on the journal webpage and are freely accessible to all readers. They include a short stand first (maximum of 300 characters, including space) as well as 2-5 one sentence bullet points that summarise the paper. Please write the bullet points to summarise the key NEW findings. They should be designed to be complementary to the abstract - i.e. not repeat the same text. We encourage inclusion of key acronyms and quantitative information (maximum of 30 words / bullet point). Please use the passive voice. Please attach these in a separate file or send them by email, we will incorporate them accordingly.

You are also welcome to suggest a striking image or visual abstract to illustrate your article. If you do please provide a jpeg file 550 px-wide x 300-800px high.

10) A Conflict of Interest statement should be provided in the main text

11) Please note that we now mandate that all corresponding authors list an ORCID digital identifier. This takes <90 seconds to complete. We encourage all authors to supply an ORCID identifier, which will be linked to their name for unambiguous name identification.

Currently, our records indicate that there is no ORCID associated with your account.

Please click the link below to provide an ORCID:

Link Not Available

Photos 400-800 DPI

*Additional important information regarding figures and illustrations can be found at

<https://bit.ly/EMBOPressFigurePreparationGuideline>. See also figure legend preparation guidelines:

<https://www.embopress.org/page/journal/17574684/authorguide#figureformat>

***** Reviewer's comments *****

Referee #1

Remarks for Author:

Is suitable for publication

The authors addressed the remaining editorial issues.

7th Feb 2024

Dear Prof. Wang,

We are pleased to inform you that your manuscript is accepted for publication and is now being sent to our publisher to be included in the next available issue of EMBO Molecular Medicine.

Yours sincerely,
